# VASCilia is an open-source, deep learning-based tool for 3D analysis of cochlear hair cell stereocilia bundles

Yasmin M. Kassim[1]*, David B. Rosenberg[1], Samprita Das[1], Xiaobo Wang[1], Zhuoling Huang[1], Samia Rahman[1], Ibraheem M. Al Shammaa[2], Samer Salim[1], Kevin Huang[1], Alma Renero[1], Yuzuru Ninoyu[3,4], Rick A. Friedman[3], Artur A. Indzhykulian[5], Uri Manor[1,3,6]*

**1** Department of Cell & Developmental Biology, University of California San Diego, La Jolla, California, United States of America, **2** Department of Cellular and Molecular Biology, University of California, Berkeley, California, United States of America, **3** Department of Otolaryngology, University of California, San Diego, La Jolla, California, United States of America, **4** Department of Otolaryngology, Kyoto Prefectural University of Medicine, Kyoto, Japan, **5** Department of Otolaryngology, Harvard Medical School and Massachusetts Eye and Ear, Boston, Massachusetts, United States of America, **6** Halıcıoğlu Data Science Institute, University of California, San Diego, La Jolla, California, United States of America

* ykassim@ucsd.edu (YMK); u1manor@ucsd.edu (UM)

## Abstract

Cochlear hair cells are essential for hearing, and their stereocilia bundles are critical for mechanotransduction. However, analyzing the 3D morphology of these bundles can be challenging due to their complex organization and the presence of other cellular structures in the tissue. To address this, we developed VASCilia (Vision Analysis StereoCilia), a Napari plugin suite that automates the analysis of 3D confocal microscopy datasets of phalloidin-stained cochlear hair cell bundles. VASCilia includes five deep learning-based models trained on mouse cochlear datasets that streamline the analysis process, including: (1) Z-Focus Tracker (ZFT) for selecting relevant slices in a 3D image stack; (2) PCPAlignNet (Planar Cell Polarity Alignment Network) for automated orientation of image stacks; (3) a segmentation model for identifying and delineating stereocilia bundles; (4) a tonotopic Position Prediction tool; and (5) a classification tool for identifying hair cell subtypes. In addition, VASCilia provides automated computational tools and measurement capabilities. Using VASCilia, we demonstrate its utility on challenging datasets, including neonatal wild type and *Eps8* KO 5-day old mice. We further showcase its power by quantifying complex bundle disorganization in *Cdh23*$^{-/-}$ cochleae via texture analysis, which revealed systematically more heterogeneous and less regular bundles than littermate controls. These case studies demonstrate the power of VASCilia in facilitating detailed quantitative analysis of stereocilia. VASCilia also provides a user-friendly interface that allows researchers to easily navigate and use the tool, with the added capability to reload all their analyses for review or sharing purposes. We believe that VASCilia will be a valuable resource for researchers studying cochlear hair cell development

**Data availability statement:** In support of open science and to enhance reproducibility, we are pleased to publicly release all annotated datasets used for training and testing in our experiments. This dataset consists of 1,870 stereocilia bundles (410 IHCs and 1,460 OHCs) from P5 697 mice, and 335 bundles (92 IHCs and 243 OHCs) from P21 mice. It is freely available to the research community, facilitating further studies, benchmarking, and advances in the field. Additionally, the complete source code for Napari plugin, has been made publicly accessible. The code can be accessed through our dedicated GitHub repository at the following link: https://github.com/ucsdmanorlab/Napari-VASCilia. The corresponding archived release is available on Zenodo at https://doi.org/10.5281/zenodo.17871652. The full documentation is accessible through this link: https://ucsdmanorlab.github.io/Napari-VASCilia/. The raw data and manual annotations used to train the final model provided with VASCilia are available at the following link: https://doi.org/10.5281/zenodo.17905676. We encourage the community to utilize these resources in their research and to contribute improvements or variations to the methodologies presented.

**Funding:** This work was supported by the Chan Zuckerberg Initiative DAF (CZI Imaging Scientist Award DOI: 10.37921/694870itnyzk), the National Science Foundation (NSF NeuroNex Award 2014862), the David F. And Margaret T. Grohne Family Foundation, the G. Harold & Leila Y. Mathers Foundation, and the National Institute on Deafness and Other Communication Disorders (NIDCD R01 DC021075-01), and the Dr. David V. Goeddel Chancellor's Endowed Chair in Biological Sciences (to U.M.). This work was also supported by NIDCD grant R018566-03S1 to U.M. and R.A.F. Microscopy in this work was supported by

and function, addressing a longstanding need in the hair cell research community for specialized deep learning-based tools capable of high-throughput image quantitation. We have released our code along with a manually annotated dataset that includes approximately 55 3D stacks featuring instance segmentation (https://github.com/ucsdmanorlab/Napari-VASCilia). This dataset comprises a total of 502 inner and 1,703 outer hair cell bundles annotated in 3D. As the first open-source dataset of its kind, we aim to establish a foundational resource for constructing a comprehensive atlas of cochlea hair cell images. Ultimately, this initiative will support the development of foundational models adaptable to various species, markers, and imaging scales to accelerate advances within the hearing research community.

## 1 Introduction

The last decade has seen a phenomenal advancement in Artificial Intelligence (AI), giving birth to countless architectures, backbones, and optimization techniques that are being continually refined [1,2]. Despite the potential to revolutionize computer vision tasks, many researchers in biological sciences still use computationally primitive and labor-intensive approaches to analyze their imaging and microscopy datasets. One major area where AI-based computer vision tasks could have maximal impact is in pre-clinical auditory research. In our lab, cutting-edge imaging tools and approaches are developed and utilized to gain insights into the structure-function relationship of subcellular structures in sensory hair cells with molecular resolution. These rich imaging datasets present an opportunity to understand how sensory hair cells function in health and disease, including cases of congenital and age-related hearing loss [3–9]. The structural attributes of the cochlea are quite uniform across species. For example, the famously snail-shaped cochlea is governed by a so-called "tonotopic" organization with a gradient of morphological features that change from one end of the cochlea to the other. The result of this organization is that each position along the length of the cochlea dictates the characteristic frequency of sound to which the tissue responds, therefore directly correlating position with function. In other words, the sensory hair cell organelles called "stereocilia" or "bundles" follow a pattern of increasing lengths as a function of position along the length of the cochlea [10,11]. The cartoon in Fig 1 illustrates how the length of hair cells corresponds to specific frequencies, when the cochlea is unrolled for illustrative purposes. Hair cells are organized tonotopically. Hair cells at the base of the cochlea respond best to high-frequency sounds, while those at the apex respond best to low-frequency sounds. The physical and functional properties of these structures vary systematically along the tonotopic axis of the auditory system. For example, hair bundles might be longer or shorter, or synapses might be more or fewer or have different properties, depending on whether they are associated with high-frequency or low-frequency regions [12,13]. The association between these attributes and their purpose makes the cochlea an ideal system for investigating the potential of AI in biological image analysis.

the Waitt Advanced Biophotonics Core of the Salk Institute with funding from the Waitt Foundation, the NIH National Cancer Institute (NCI CCSG P30 014195), and the San Diego Nathan Shock Center (P30AG068635). This work was also supported by NIH NIDCD R01DC020190, R01DC017166 and R01DC021795 to A.A.I. The funders had no role in study design, data collection and analysis, decision to publish, or preparation of the manuscript.

**Competing Interests:** The authors have declared that no competing interests exist.

In this study, we leverage Napari, an open-source Python tool [14], as the foundation for developing our plugin, chosen for its robust viewer capabilities. The Napari platform has seen a growing number of plugins [15–28] designed to address various biomedical challenges. However, our plugin represents the first such tool tailored specifically for the ear research community, enabling high-precision AI-based 3D instance segmentation and measurement quantification of stereocilia cells.

Previous studies have explored manual segmentation of stereocilia images to quantify measurements for specific research objectives [9,29]. Others have attempted to automate this process using traditional intensity-based methods [30]. However, these traditional approaches typically lack the capabilities provided by AI technologies, which can autonomously extract features and identify regions without relying exclusively on intensity cues. This gap underscores the critical need for advanced AI-driven segmentation techniques that utilize a broader array of features, thereby enhancing both the accuracy and the detail of the segmentation. We identified a few machine learning and deep learning publications. Urata et al. [31] relies on template matching and Machine learning-based pattern recognition to achieve detection and analysis of hair cell positions across the entire longitudinal axis of the organ of Corti. Buswinka et al. [32] develops a software tool that utilizes AI capabilities; however, this tool is limited to providing bounding box detection and does not offer 3D instance segmentation for the bundles. This significantly hampers the tool's ability to provide shape-level descriptors and accurate measurements. Cortada et al. [33] use stardist deep learning architecture [34] to segment the hair cells, however, they don't provide 3D instance segmentation, they only provide 2D detection based on max projection.

VASCilia (Fig 18) provides a comprehensive workflow assisted by AI for automated 3D hair cell image analysis: 1. Read and pre-processes 3D image *z*-stacks. 2. Removes out-of-focus frames (AI-assisted). 3. Aligns stacks to the planar cell polarity (PCP) axis (AI-assisted). 4. Generates instance-labeled 3D segmentation masks for downstream analysis (AI-assisted). 5. Computes 2D/3D measurements (e.g., volume, centroid, surface area) and additional derived metrics. 6. Measures tip-to-base height for each hair-cell bundle. 7. Classifies bundles into four rows (IHC, OHC1, OHC2, OHC3) (AI-assisted). 8. Quantifies protein intensity to assess expression levels. 9. Estimates bundle orientation relative to the PCP axis. 10. Identifies the cochlear region of origin (base, middle, apex) using a pre-trained model (AI-assisted). 11. Supports model fine-tuning on user data to accommodate staining protocols and experimental conditions (AI-assisted). 12. Enables full export/import of the analysis state (data and intermediates) via a single pickle file for reproducibility.

To demonstrate the robustness and practical utility of VASCilia, we specifically apply it to challenging preclinical datasets. These include neonatal (P5) mice, whose shorter stereocilia are historically difficult to analyze, and feature a range of phenotypes from wild type (WT) to knockout (Eps8 KO) and AAV-rescued genotypes. Furthermore, we showcase VASCilia's ability to move beyond simple measurements by quantifying complex bundle disorganization in a genetic model of deafness (*Cdh23* mutants), proving its capability to analyze intermediate and abnormal phenotypes that are otherwise intractable to quantify at scale. We anticipate that our AI-enhanced

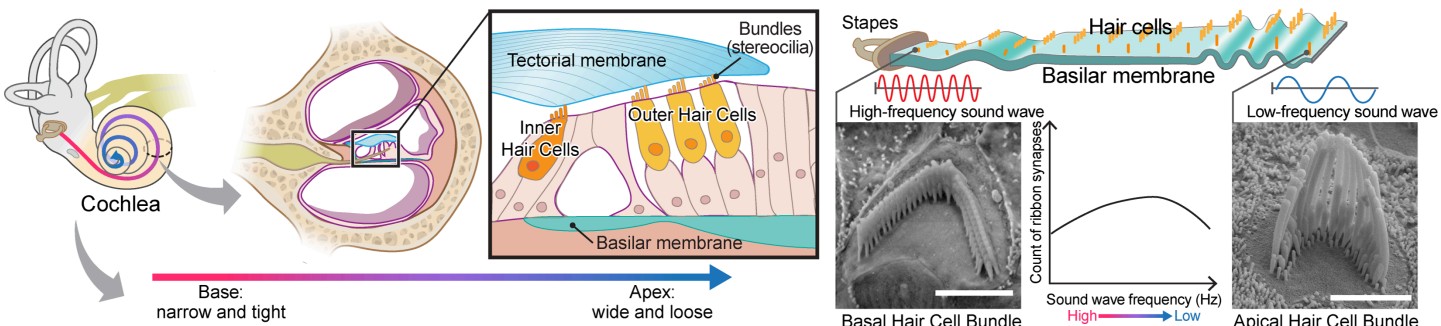

**Fig 1. The anatomy of the cochlea [35].** Sensory hair cell stereocilia sit along a tonotopic axis that follows the length of the spiral-shaped cochlea with a striking pattern of increasing stereocilia lengths from the base to the apex of the cochlea. The morphological and spatial features of cochlear hair cell stereocilia follow extremely predictable tonotopic patterns between individuals and species: Stereocilia lengths increase as a function of position along the tonotopic axis of the cochlea, which in turn is reflective of the frequency of sound they are tuned to detect. Thus, the relationship between the morphological and spatial features of these cells and their function are relatively well-defined compared to many other biological systems. Due to their highly patterned organization, cochlear tissues thus present a particularly striking opportunity for automated computer vision tasks.

plugin will streamline the laborious process of manual quantification of hair cell bundles in 3D image stacks, offering high-throughput and reproducible analysis with clear biological interpretation.

## 2 Results

### 2.1 Z-Focus tracker for 3D images of cochlear hair cells

In this section, we present the performance and evaluation of our custom network, ZFT-Net, along with comparisons to other commonly used architectures such as ResNet10, ResNet18, DenseNet121, and EfficientNet. Our goal was to classify image frames within each 3D z-stack into three zones: the pre-cellular zone (PCZ), the cellular clarity zone (CCZ) and the noise saturation zone (NSZ) to focus segmentation and analysis on the portion of the z-stack that contains clear cellular structures. The results shown in Table 1 and Fig 2 highlight the performance of the networks in metrics such as accuracy, precision, recall, specificity, F1 score and error rate, and prediction stability. We found ZFT-Net produced the best results. All metrics were averaged across all three classes and calculated using true positives (TP), false positives (FP), false negatives (FN), and true negatives (TN) derived from the confusion matrix.

One notable feature of ZFT-Net is that it does not produce prediction fluctuation errors, which we named "switches". This consistency in its predictions offers a high degree of confidence, making ZFT-Net our primary choice for VASCilia. As described in Sect 4.3, the preparation of the data and the configuration of CNN played a crucial role in achieving these results. We speculate that ZFT-Net outperforms the other well-known networks because our images are downsampled to 256x256 resolution. Deeper networks tend to lose spatial characteristics in the process, which negatively impacts their performance. The architecture of ZFT-Net, on the other hand, is better suited to preserve these critical spatial features, resulting in better performance across all metrics.

$$\text{Accuracy} = \frac{TP+TN}{TP+TN+FP+FN} \quad \text{Precision} = \frac{TP}{TP+FP} \quad \text{Recall} = \frac{TP}{TP+FN} \quad \text{Specificity} = \frac{TN}{TN+FP}$$

$$\text{F1-Score} = 2 \times \frac{\text{Precision} \times \text{Recall}}{\text{Precision} + \text{Recall}} \quad \text{Error Rate} = \frac{\sum_{i \neq j} \text{Confusion Matrix}_{i,j}}{\sum_{i,j} \text{Confusion Matrix}_{i,j}}$$

**Table 1**. **Comparison of architectures based on performance metrics for Z-focus tracker.** The results highlight that ZFT-Net outperforms all other architectures across all evaluation metrics. Notably, ZFT-Net achieved a switch count of 0, meaning its predictions consistently transitioned from class 0 (pre-cellular zone) to class 1 (cellular clarity phase), and finally to class 2 (noise saturation zone), without fluctuations or misclassifications. This indicates the model's robustness in maintaining a stable and accurate prediction sequence. Source data are provided in *Supplementary Information S1 Data.xlsx*.

| Architecture | Accuracy | Precision | Recall | Specificity | F1 Score | Error Rate | Confusion Matrix | Switches |
|---|---|---|---|---|---|---|---|---|
| ZFT-Net | **97.66** | **94.14** | **96.01** | **98.48** | **94.97** | **0.03** | 237 1 0 / 6 72 7 / 0 3 159 | **0** |
| ResNet18 | 94.36 | 89.59 | 87.46 | 95.81 | 88.16 | 0.08 | 236 5 0 / 6 65 23 / 1 6 143 | 9 |
| ResNet10 | 94.77 | 92.18 | 88.65 | 95.89 | 89.52 | 0.07 | 231 3 0 / 12 72 22 / 0 1 144 | 6 |
| DenseNet121 | 95.87 | 92.43 | 90.67 | 97.05 | 91.21 | 0.06 | 241 5 1 / 2 69 20 / 0 2 145 | 3 |
| EfficientNet | 95.46 | 90.01 | 89.98 | 96.68 | 89.99 | 0.06 | 236 9 0 / 7 60 10 / 0 7 156 | 3 |

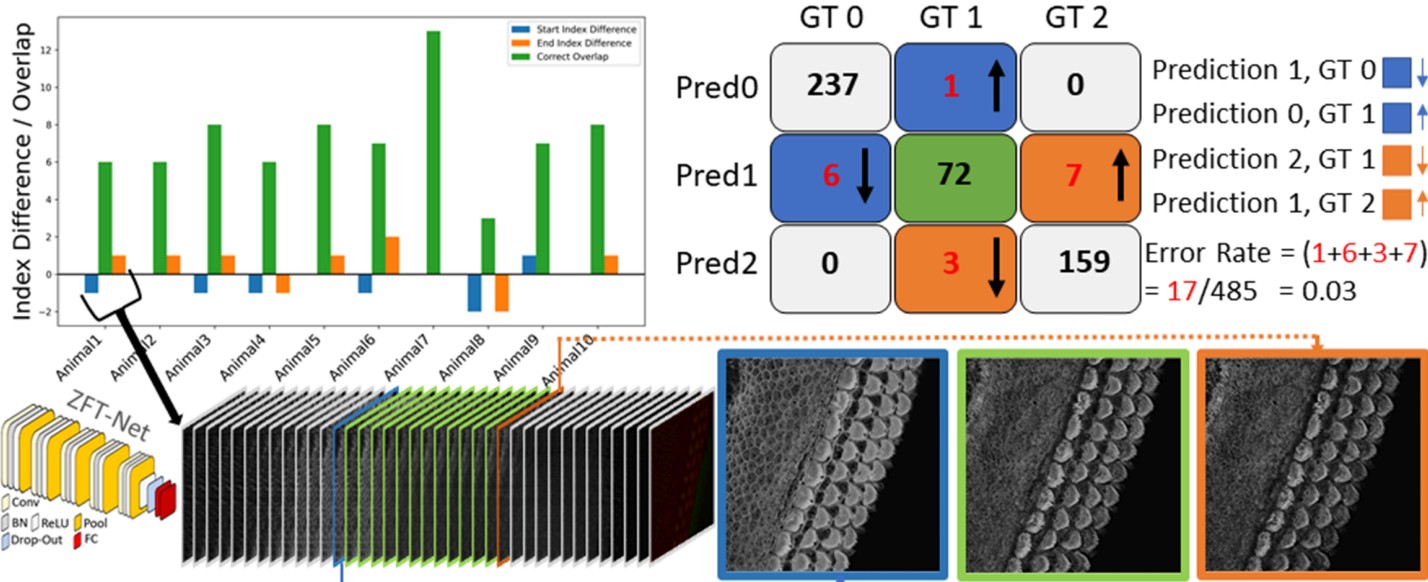

**Fig 2**. **The confusion matrix for ZFT-Net is presented alongside a corresponding bar plot, illustrating the overlapping predictions for the CCZ class, represented by the digit 1.** Errors are color-coded and oriented to reflect the different cases in the confusion matrix. GT0, GT1, and GT2 correspond to the Pre-Cellular Zone (PCZ), Cellular Clarity Zone (CCZ), and Noise Saturation Zone (NSZ), respectively. At the bottom of the figure, we provide an example of the first subject's prediction, visualized through a barplot. It shows a smooth transition in prediction from 0 (PCZ), to 1 (CCZ), to 2 (NSZ) without any fluctuation errors, except for two minor boundary errors that are likely due to annotator subjectivity. For ZFT-Net, `Conv` represents Convolution layers, `BN` stands for Batch Normalization, `ReLU` is the Rectified Linear Unit activation function, `Pool` refers to Pooling layers, and `FC` denotes Fully Connected layers.

## 2.2 Rotation correction and alignment of 3D image stacks with PCPAlignNet

The orientation of 3D cochlear image stacks is highly variable due to inconsistencies in imaging parameters, tissue handling, and the complex spiral structure of the cochlea. Automated orientation of cochlear hair cell image stacks can significantly accelerate batch processing, which is essential for efficiently handling large datasets. To address this challenge, we generated training data for PCPAlignNet, which aligns each 3D stack to orient the hair cell bundles along the planar polarity axis.

After data preparation and augmentation Sect 4.4, our training dataset resulted in 72 folders (classes), each containing 1,831 images. We experimented with several networks, and DenseNet121 yielded the best performance; see Table 2.

**Table 2**. **Performance comparison of different architectures using various error metrics reveals that DenseNet121 has optimal performance across all the metrics.** Source data are provided in *Supplementary Information S1 Data.xlsx*.

| Architecture | MAE | RMSE | Max Error | Median Absolute Error | R-squared |
|---|---|---|---|---|---|
| ResNet50 | 10.53 | 41.36 | 180 | 0 | 0.86 |
| ResNet18 | 10 | 41.33 | 180 | 0 | 0.86 |
| MobileNet | 10 | 41.33 | 180 | 0 | 0.86 |
| DenseNet121 | **0** | **0** | **0** | **0** | **1** |

During inference, we corrected the orientation of the stereocilia bundles by predicting the rotation angle for each image in the tested stack using the pre-trained model. For each frame, the model produced class scores corresponding to possible angles of rotation. The predicted scores were averaged across all frames in the stack to ensure that the final decision accounted for the model's predictions for every frame. The angle corresponding to the highest average score (class × 5°) was selected as the final predicted orientation. To achieve proper alignment, the images were rotated by 360° minus the predicted angle.

### 2.3 A 3D instance segmentation pipeline for stereocilia bundles

Our method utilizes 2D detection on individual frames, which is then followed by reconstructing the 3D object through a multi-object assignment algorithm. For detailed methodology, refer to Sect 4.6.

We partitioned the dataset into training, testing, and validation groups at the stack level to avoid data leakage and mixing frames from different stacks. The training set includes 30 stacks, while the validation contains 5 stacks. Ten stacks are withheld for testing and evaluation. These stacks are categorized into two types: typical and complex cases. Six stacks fall into the typical category, featuring images that are easier to segment due to less overlap among the bundles and well-separated bundle rows. The remaining four stacks are classified as complex cases. These contain structures that pose significant challenges for segmentation due to various factors. The objective of this partitioning is to evaluate the decrease in performance when processing challenging cases and check the robustness of the detection algorithm. Fig 3A–3E illustrate the model's proficiency in accurately detecting the cells and demonstrate the model's effectiveness in overcoming the inherent challenges associated with this task. As a representative single-stack example, Fig 4 shows 13 frames from one stack segmented in 2D; a multi-object assignment algorithm then reconstructs each 3D object. Details are provided in Sect 4.6.

After completing the training process, we employed the trained model to segment each bundle in the images from the testing set. To evaluate performance, we calculate the Intersection over Union (IoU) for 3D volumes by comparing predicted and ground truth segmentation masks. It is defined as $IoU = \frac{|A \cap B|}{|A \cup B|}$, where $|A \cap B|$ represents the volume of the intersection of the two volumes, and $|A \cup B|$ represents the volume of their union. In addition, we calculate the F1 measure, accuracy, precision, and recall for the predictions against the ground truth. The evaluation process iterates through unique labels found in both ground-truth and predicted volumes, excluding the background label, and computes the IoU for the corresponding labels. This method tracks TP, FP, and FN across varying IoU thresholds. For instance, if one 3D bundle in the ground truth overlaps with two bundles in the predictions, the algorithm considers the one with the largest overlap as a TP and the unmatched one as a false positive. We perform evaluations across a range of IoU thresholds, from 0.1 to 1 in steps of 0.05. In object detection, we do not consider the True Negative (TN) because it refers to correctly labeling background pixels or non-object pixels as such. Since the background or non-object areas can be vast, focusing on TNs would skew the performance metric towards the most common class (background), which does not provide useful information about the model's ability to detect objects of interest.

Fig 5A indicates that the average F1-measure and accuracy are 99.4% and 98.8%, respectively, at an IoU of 0.5 for typical test set. In contrast, Fig 5C reveals that at the same IoU level, the average F1-measure and Accuracy decrease to

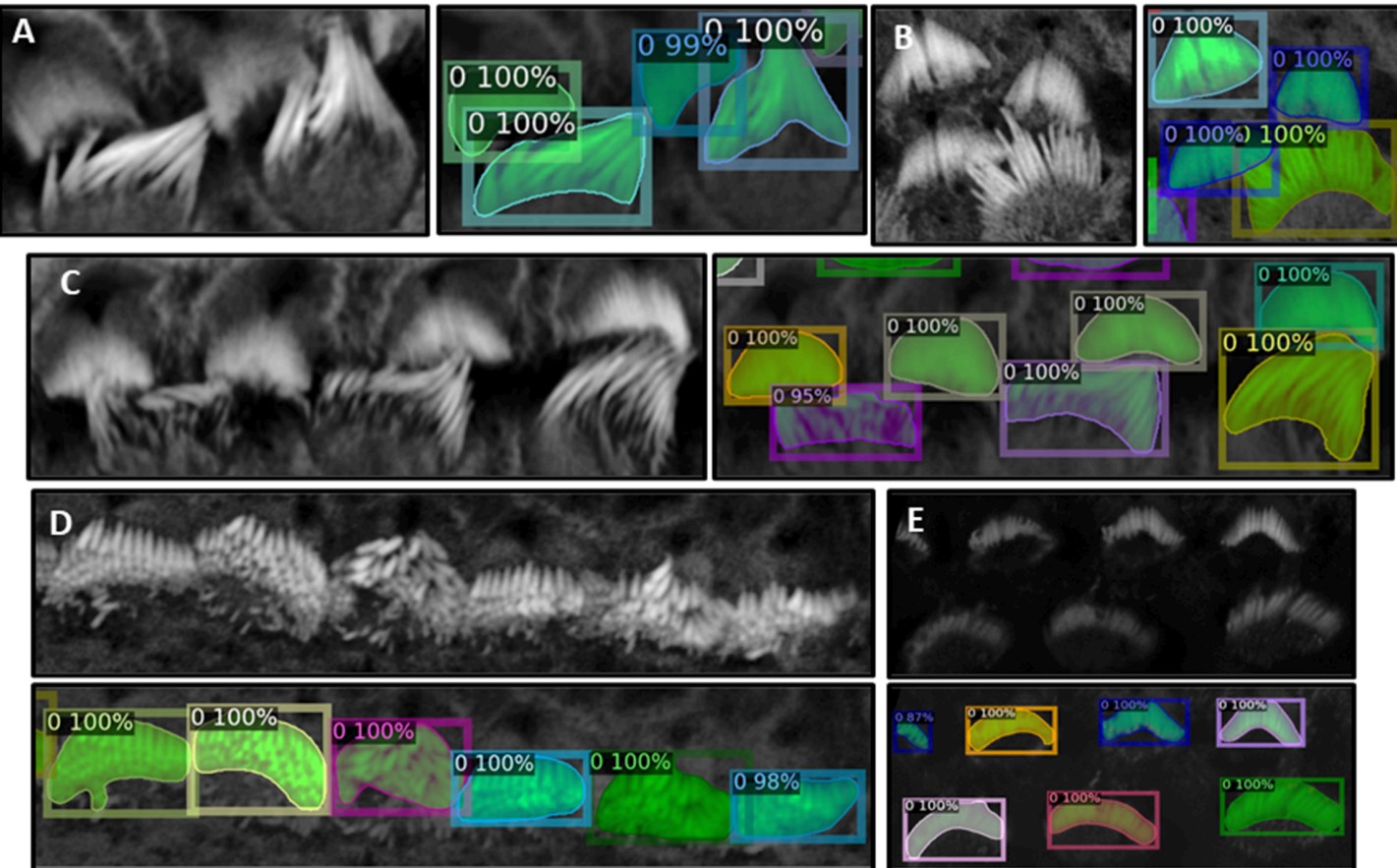

**Fig 3**. **Panels A, B, and C show instances where inter-row stereocilia bundles are in close proximity, yet the algorithm successfully separates them, demonstrating robust performance. Panel D** presents a scenario where intra-row bundles are tightly clustered; however, the algorithm efficiently distinguishes each bundle, highlighting its ability to resolve complex spatial relationships within the samples. In **Panel E**, despite the raw image being notably dark, the algorithm remains effective in detecting and segmenting individual bundles.

95.7% and 91.9% for complex test set. Maintaining the F1 measure and accuracy above 90% demonstrates that the algorithm effectively handles the significant challenges present in the images. We can observe from Fig 5B and 5D that the total number of TPs remains high until the IoU reaches 0.8 and 0.72, respectively. Beyond these thresholds, there is a noticeable increase in both FPs and FNs. When you observe that TPs remain high until IoU thresholds of 0.8 and 0.72, this indicates that the algorithm is quite effective at correctly identifying and matching relevant objects or features in the dataset up to these points. The increasing errors (FPs and FNs) at higher IoU thresholds indicate that the algorithm struggles with precision and recall balance as criteria become more strict. Fig 6 showcases ten crops of stereocilia bundles: the first row displays the raw crops, the second and third rows features ground truth manual annotations made by two human annotators in the open-source Computer Vision Annotation Tool (CVAT [36]), and the fourth row presents our predicted 3D volumes for each bundle. It is evident that most bundles achieve IoU scores between 0.7 and 0.8, which visually suggest near-perfect alignment.

PLOS Biology

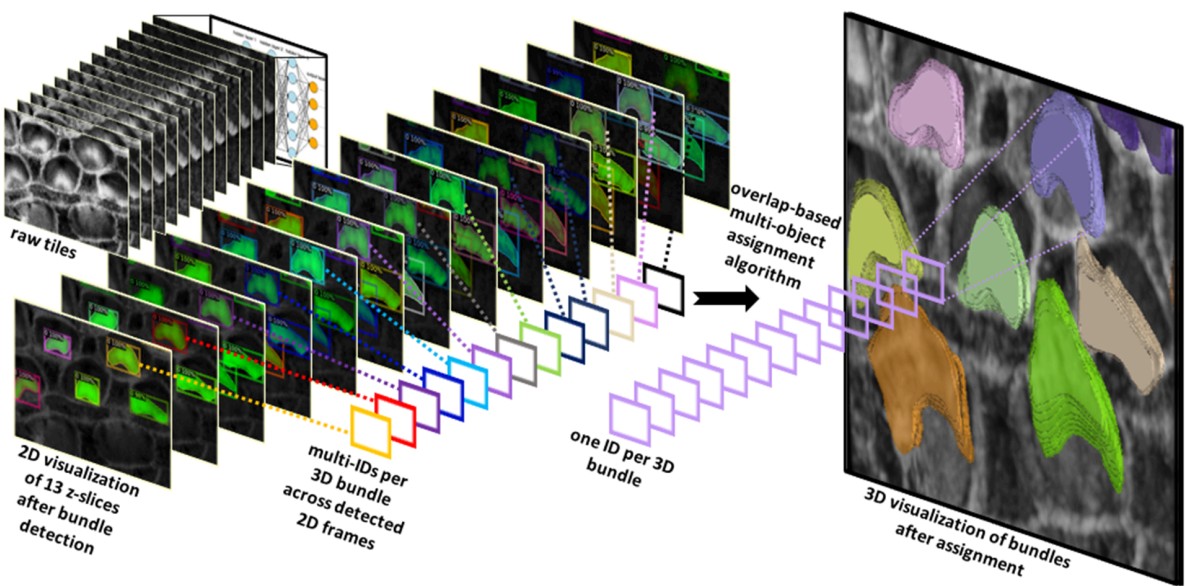

**Fig 4**. **The algorithm successfully detects stereocilia bundles in 2D across successive frames, assigning unique IDs to each detected object within the 2D plane.** The variation in IDs across frames reflects the independent detection process in each 2D frame. Subsequently, a multi-object assignment algorithm intervenes to reconcile these IDs, effectively re-assigning them to maintain consistency across frames. This step is crucial for reconstructing accurate 3D objects, ensuring that each bundle retains a consistent ID throughout all frames.

The dataset has been annotated by five different human annotators. To put the accuracy of our predictions in a more meaningful context, we investigated the margin of error between annotators on the same stack. We found that the average IoU for 47 instances from a single stack between two annotators is 0.70. Strikingly, the model achieved a higher average IoU of 0.76 with the first annotator and 0.74 with the second annotator. We speculate this is due to the model's ability to effectively average the experience of multiple annotators (five annotators in our case). These results also highlight the difficulty of achieving accuracies beyond a certain amount, as even two human expert annotators only agreed 70% on pixel-based 3D annotations. In addition, Paired t-tests revealed that the model's predictions are significantly better aligned with both Annotator 1 ($t = 3.7399$, $p = 0.0002$) and Annotator 2 ($t = 7.8134$, $p = 0.0000$) compared to the agreement between the two annotators. No significant difference was found between the model's alignment with Annotator 1 and Annotator 2 ($t = -1.7431$, $p = 0.0879$), indicating that the model generalizes equally well across both sets of annotations (Fig 7A and 7B). We speculate that the model may be averaging the annotators' varying styles effectively, and that these results underscore the robustness of the model in producing reliable and consistent predictions, which, in some cases, surpass the level of agreement between human expert annotators.

## 2.4 VASCilia computational tools and measurements

The Napari plugin equips users with essential measurements and deep learning-based tools tailored for analyzing cochlear hair cell stereocilia bundles. Users can accurately obtain up to 15 different 2D and 3D measurements, including volume, surface area, and centroids of segmented regions. Beyond these fundamental measurements, the plugin includes specialized metrics critical for hair cell research, such as calculating stereocilia bundle heights, predicting the region of the cochlea from which the stack is taken (Base, Middle, Apex), classifying hair cells (e.g. inner vs. outer hair cell subtypes), measuring fluorescence signal within each bundle, and determining bundle orientation with respect to the planar polarity axis of the cochlea. These features are designed to support the most common goals of the hair cell imaging research community.

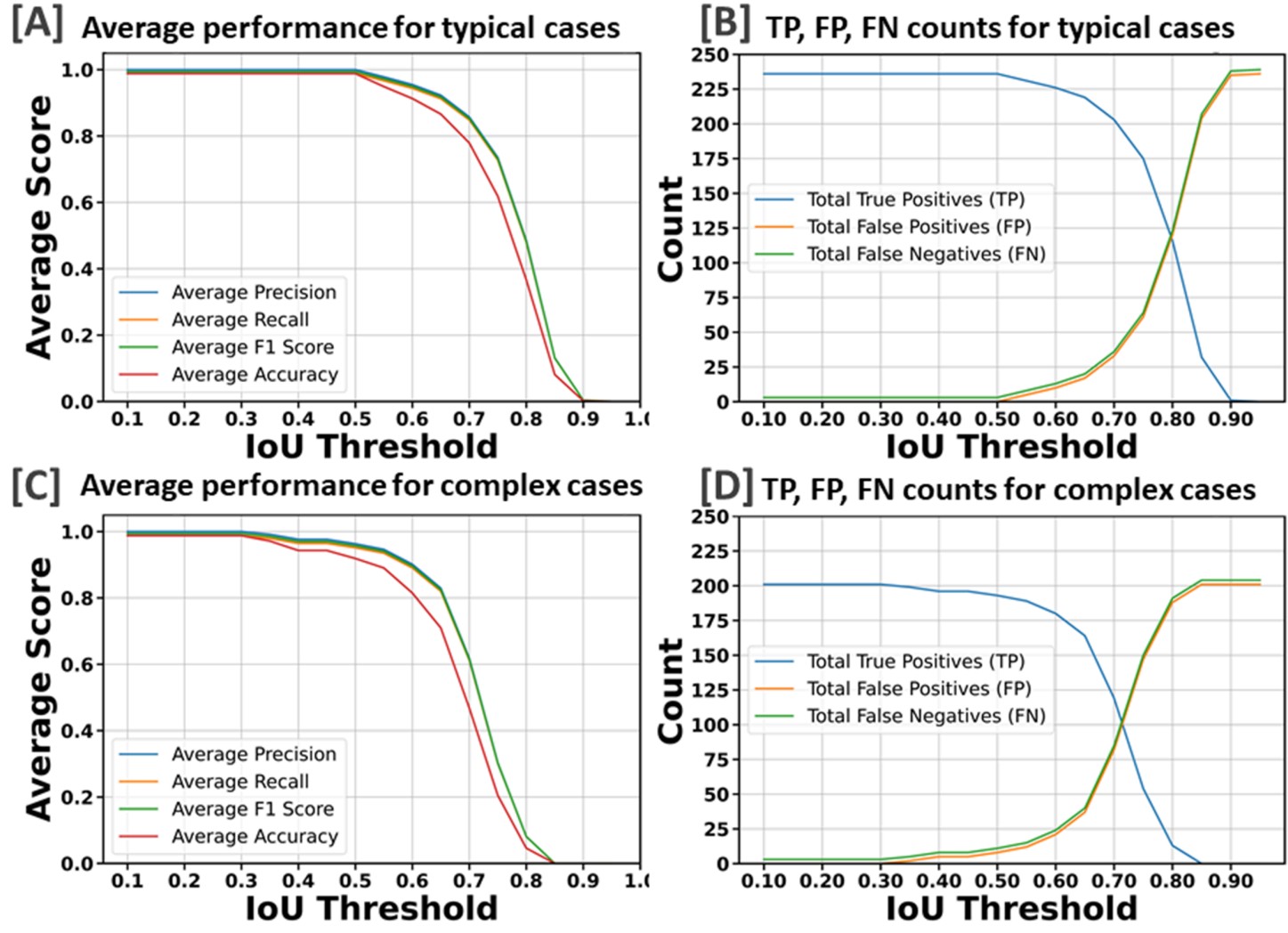

**Fig 5**. **Detailed performance indicators are presented.** The metrics are evaluated across two categories: typical cases that do not present extreme challenges and complex cases affected by factors such as contrast variations, noise, overlapping bundles, and close proximity between them. Source data are provided in *Supplementary Information S1 Data.xlsx*.

### 2.4.1 Validation of Stereocilia bundle height measurement.

VASCilia significantly reduces the time required for bundle height measurements. The manual and computational approaches used for these measurements are detailed in Sect 4.7. The plugin provides users with a CSV file that details the height measurements for each bundle, each tagged with a corresponding ID. Fig 8 illustrates the computation and Table 3 and Fig 9 present bundle height validation as measured between VASCilia and two human expert annotators for 18 cells from four different datasets (two WT and two *Eps8* KO with AAV injection [37]). We found it takes an average of 5.5 minutes to manually annotate each bundle height using the most commonly used open-source microscope image analysis tool, Fiji [38] In contrast, VASCilia significantly reduces this time to just one second to press a button to calculate the height of all the bundles in a single 3D stack (as many as 55), with at most five minutes needed for refining the base and top points of stack bundles if necessary. Thus, measuring the length for 10 stacks, each containing 50 cells, would take 2,750 minutes (approximately 46 hours) with Fiji, compared to only 50 minutes with VASCilia. We conducted both paired t-tests and Pearson correlation analyses to evaluate the

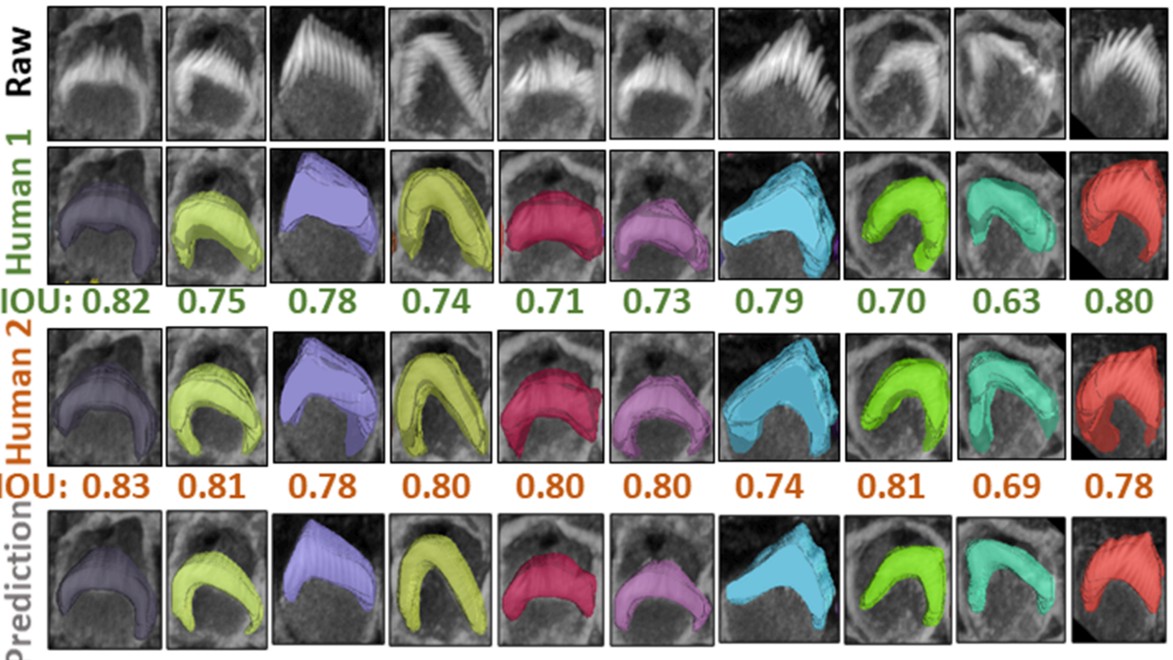

**Fig 6**. **Comparison of raw crops (first row), GT masks for the first human annotator (second row), GT masks for the second human annotator (third row), and the Predicted masks (fourth row) for 10 different crops from the same stack, the IoU score underneath the crops indicate the score between the 3D GT masks and the 3D predicted masks.** We examined the margin of error between two annotators for a single 3D stack over 47 instances. The average overlap was 0.70 between two annotators, 0.74 between one annotator and the prediction, and 0.76 between the second annotator and the prediction.

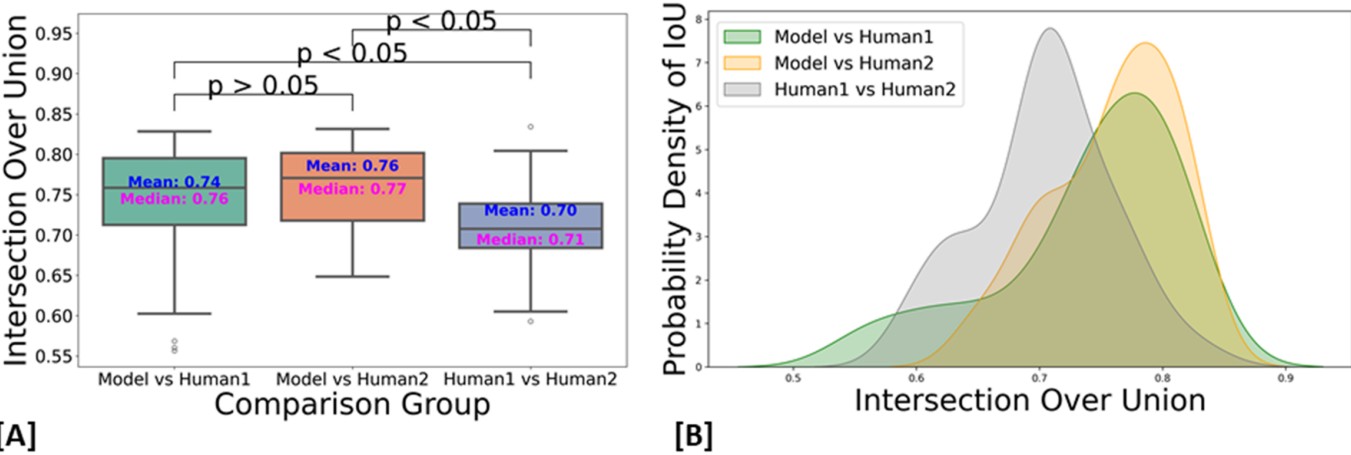

**Fig 7**. **Comparison of the IoU distribution (left) and density plot (right) between model predictions and annotators. Source data are provided in *Supplementary Information S1 Data.xlsx*.** [A] Box plot comparing the IoU alignment of model predictions with Human 1 and Human 2, and the agreement between Human 1 and Human 2. Statistical tests indicate that the model predictions are significantly better aligned with both Human 1 (p = 0.0002) and Human 2 (p = 0.0000) compared to the agreement between the two annotators. Notably, there is no significant difference between the model's alignment with Human 1 and Human 2 (p = 0.0879). [B] A density plot (KDE) estimates the distribution of IoU values by smoothing the data with small curves (kernels) at each point. The y-axis shows the density, representing the relative likelihood of observing the IoU values at different points on the x-axis. The density curves for Model vs. Human 1 and Model vs. Human 2 show considerable overlap, indicating similar IoU distributions between the model and both annotators. In contrast, Human 1 versus Human 2 shows less overlap, suggesting greater variability between the annotators' manual segmentation.

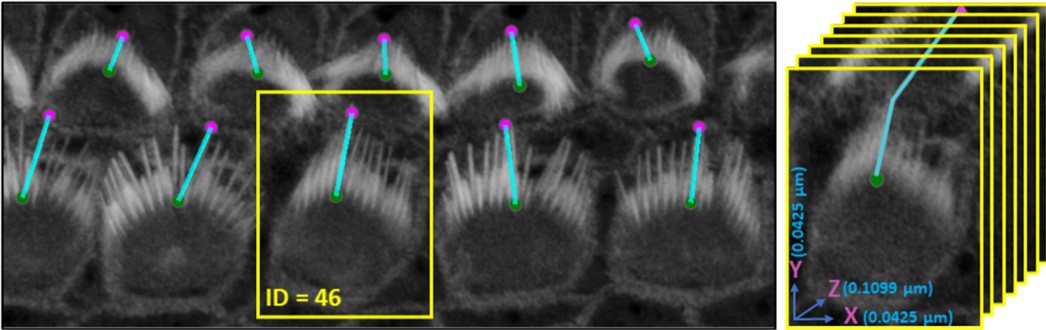

**Fig 8**. **An illustration of the plugin's automated method for measuring the distance from the tip to the bottom of stereocilia bundles.** Users can adjust the positions of the upper and lower points. Upon making these adjustments, the plugin's listeners automatically detect the changes, redraw the connecting line, and recalculate the distance accordingly. Left: shows the 3D visualization using Napari. Right: demonstrates how the distance is computed from x, y, and z, considering the physical resolution in microns.

**Table 3**. **Measurements from VASCilia and Observers with Descriptive Statistics**

| Sample | ID | VASCilia | Observer1 | Observer2 |
|---|---|---|---|---|
| DataSet1 | 2 | 3.520 | 3.513 | 3.431 |
| | 19 | 2.786 | 2.675 | 2.883 |
| | 25 | 4.129 | 3.590 | 3.947 |
| | 32 | 3.728 | 3.367 | 3.613 |
| DataSet2 | 10 | 3.816 | 3.474 | 3.466 |
| | 32 | 3.028 | 2.633 | 2.454 |
| | 24 | 3.748 | 3.534 | 3.575 |
| | 31 | 3.572 | 3.389 | 3.159 |
| DataSet3 | 3 | 2.006 | 2.189 | 2.646 |
| | 39 | 1.678 | 1.897 | 2.426 |
| | 34 | 2.099 | 2.202 | 2.006 |
| | 46 | 4.314 | 4.543 | 4.313 |
| | 33 | 3.671 | 3.624 | 2.944 |
| | 48 | 3.304 | 3.519 | 3.194 |
| DataSet4 | 3 | 2.738 | 2.446 | 2.916 |
| | 47 | 2.797 | 2.706 | 3.206 |
| | 17 | 2.531 | 2.628 | 2.044 |
| | 54 | 3.004 | 2.559 | 1.965 |
| **Statistics** | | **Mean** | **Median** | **Std. Dev.** |
| VASCilia | | 3.137 | 3.166 | 0.725 |
| Observer 1 | | 3.027 | 3.037 | 0.663 |
| Observer 2 | | 3.010 | 3.052 | 0.649 |

agreement between VASCilia and human expert annotators. The Pearson correlation analysis showed very strong positive relationships: 0.942 (p < 0.001) between VASCilia and annotator 1, 0.802 (p < 0.001) between VASCilia and annotator 2, and 0.830 (p < 0.001) between annotators 1 and 2. The paired t-test results indicate no statistically significant differences between the measurements obtained by VASCilia and annotator 1 (t = 1.868, p = 0.078) or annotator 2 (t = 1.191, p = 0.249). No significant difference (t = 0.180, p = 0.860) was observed between the measurements from observers 1 and 2. Together, our findings suggest VASCilia performs comparably to human expert annotators, and that the annotators themselves are consistent in their measurements, see Table 4.

To further validate stereocilia–length quantification, we compared VASCilia against manual Fiji measurements on $n = 15$ *Eps8*-KO cells sampled from six stacks (two Base, two Middle, two Apex) drawn from the dataset used in Fig 10A. We also evaluated $n = 15$ cells lacking functional *Cdh23* [39] (*Cdh23*$^{v-6J/v-6J}$, referred to as *Cdh23*$^{-/-}$), chosen

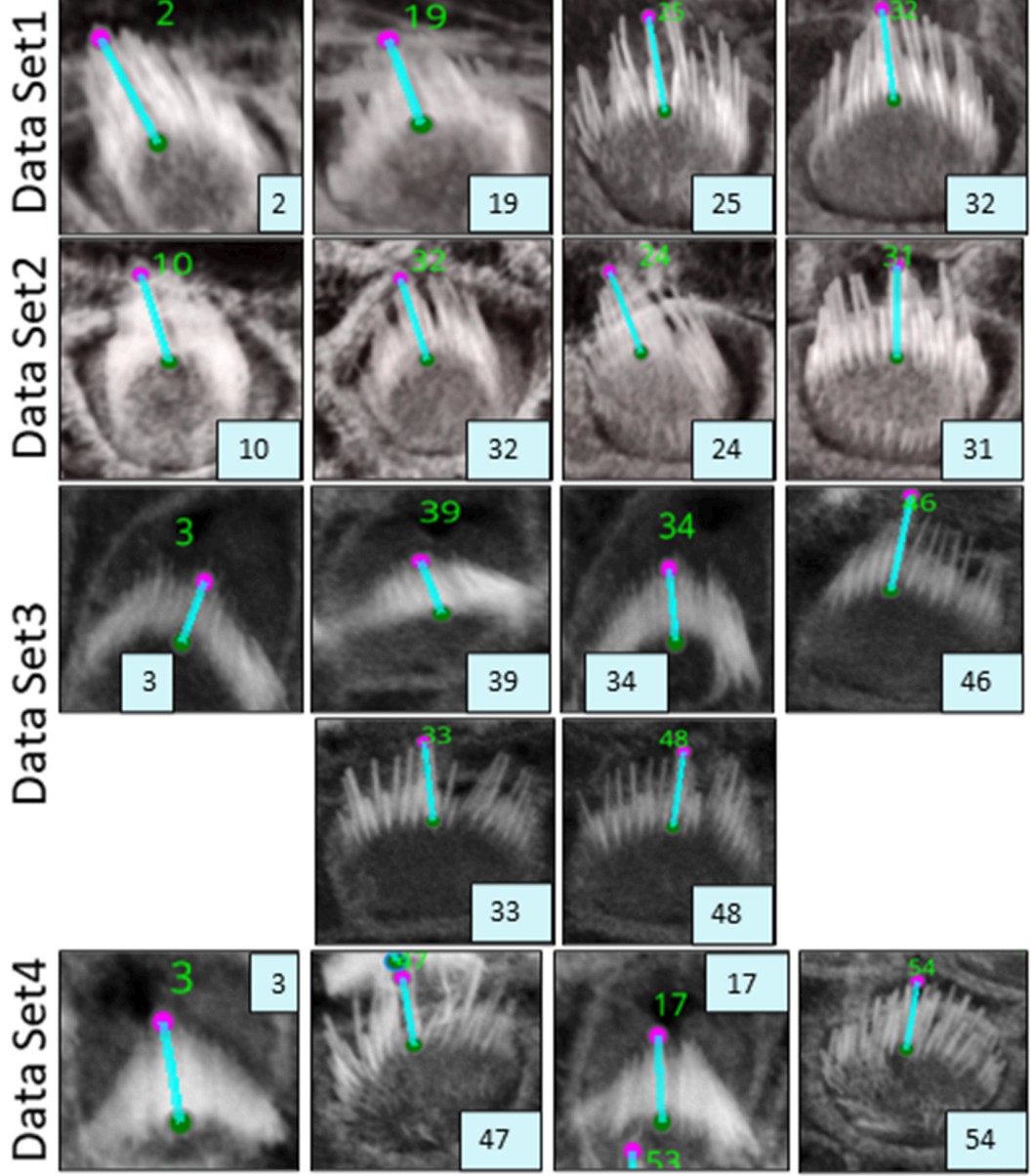

**Fig 9**. Data set crops that are associated to the computations used in Table 3, See Fig S2 A-D, where all four complete datasets are visualized for reference.

**Table 4**. Pearson correlation coefficients, p-values, and paired t-test results.

| Comparison | Correlation Coefficient | Correlation p-value | t-test Statistic | t-test p-value |
|---|---|---|---|---|
| VASCilia vs Observer 1 | 0.942 | <0.001 | 1.868 | 0.078 |
| VASCilia vs Observer 2 | 0.802 | <0.001 | 1.191 | 0.249 |
| Observer 1 vs Observer 2 | 0.830 | <0.001 | 0.180 | 0.860 |

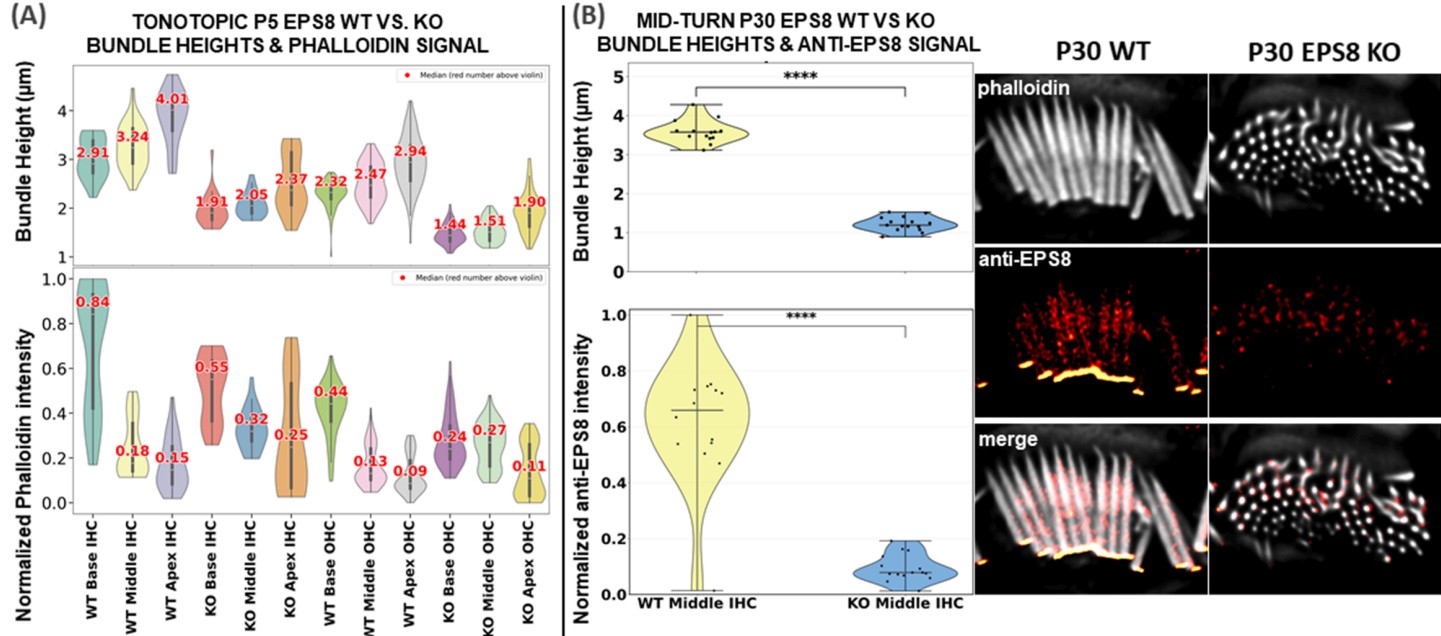

**Fig 10**. **This figure consists of two panels.** Panel (A) on the left shows bundle heights and tonotopic variations of phalloidin fluorescence intensity in P5 mouse cochlear hair cells across all cochlear regions. The tonotopic gradient observed here may not be representative of other developmental ages, which were not examined in this study. Panel (B) presents the anti-EPS8 analysis, comparing anti-EPS8 fluorescence intensity between wild type and knockout stereocilia bundles from the mid-cochlear region at P30. Source data are provided *Supplementary Information S1 Data.xlsx*.

to represent significantly disrupted bundles to assess generalization. Across both cohorts, the mean lengths were highly comparable and the paired differences were not statistically significant (paired *t*-test and Wilcoxon signed-rank); see S7 and S8 Tables.

**2.4.2 Fluorescence intensity measurements.** Understanding signal levels whether they are one or multiple proteins or a particular stain is crucial in various research fields, including cochlear function. The plugin enables the user to get a precise quantification for the signal to allow researchers to gain deeper insights into its function and potential involvement in hearing loss.

The plugin has a mechanism to calculate both the mean and total fluorescence intensity, subtracting the background intensity (optional) to correct for ambient noise. This is achieved by superimposing the 3D segmented masks onto the corresponding fluorescent image slices and aggregating the fluorescent intensities across the z-stack.

The resulting intensities are normalized using max normalization and plotted, allowing for comparative analysis across different cell types such as inner hair cells (IHCs) and outer hair cells (OHCs) in various categories (OHC1, OHC2, and OHC3). Data for each cell type is stored and visualized through histogram that display the mean and total intensity distributions to provide insights and observation about the protein or signal expressions across the cochlear architecture. To enable further downstream analysis, users will obtain plots for each cell type, resulting in a total of 20 outputs per stack (10 plots and 10 CSV files). See Fig 11 for the total intensity histogram bars for all cells in three modes, illustrated in Sect 3, from the Napari plugin. The plugin can also be used to analyze and generate plots for other proteins in different channels as needed by the study.

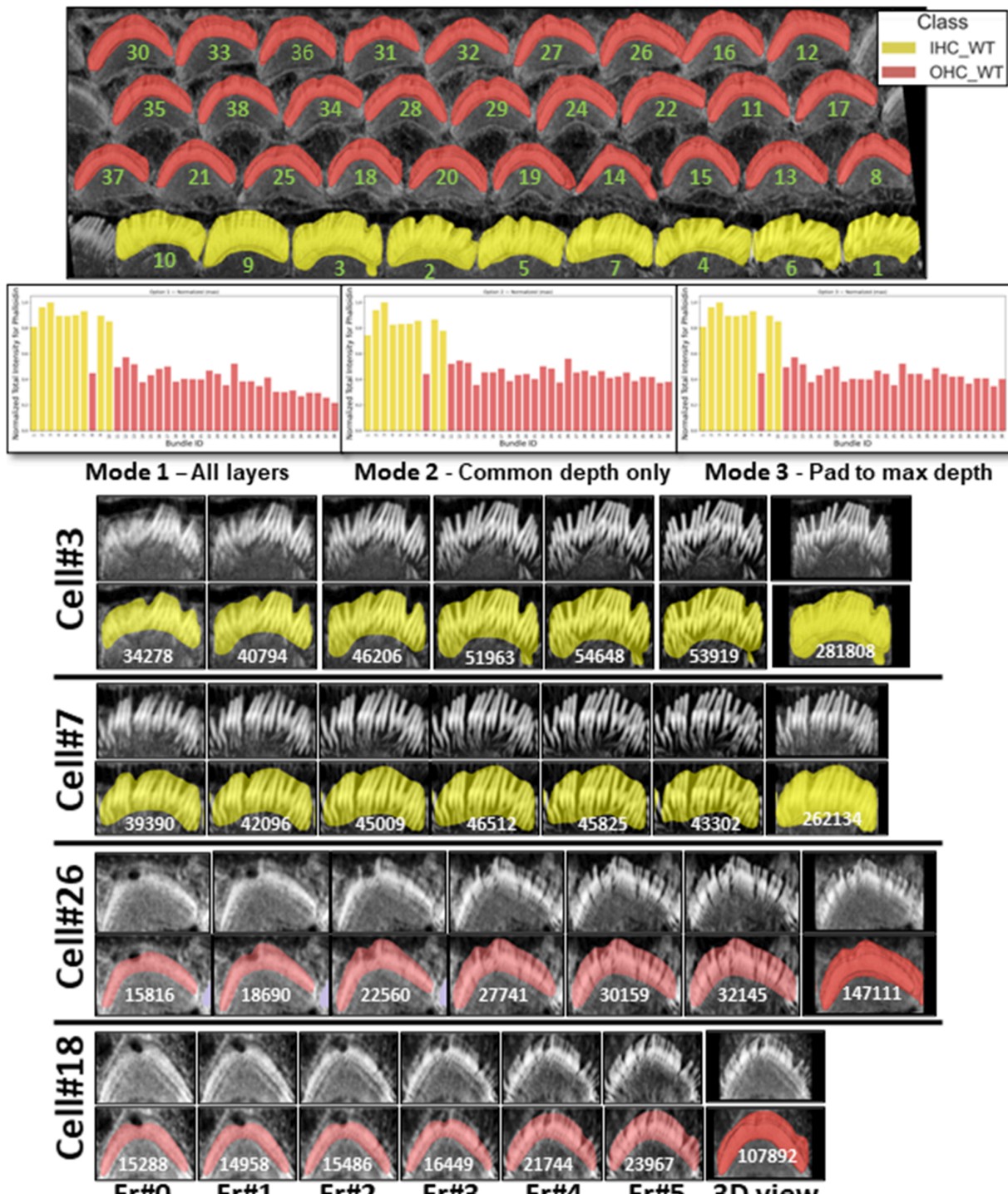

**Fig 11**. The plot demonstrates the utility and ease with which users can compare data between bar plots and actual cellular images within the plugin. **Top:** Segmented bundles; class legend (IHC WT = yellow, OHC WT = red). Green numbers mark each bundle's ID. **Middle:** Per-bundle fluorescence (phalloidin) for three depth-aggregation modes: **Mode 1:All layers** (use each bundle's full depth); **Mode 2: Common depth only** (truncate to $n$ = min depth across bundles); **Mode 3: Pad to max depth** (pad shallower bundles to $n$ = max depth by repeating the weakest-layer value, then sum). **Bottom:** Galleries from four example cells (shared same depth). Across cells, IHC bundles show consistently higher fluorescence than OHC bundles, reflecting greater F-actin signal. Source data are provided in *Supplementary Information S1 Data.xlsx*.

### 2.4.3 Case Study 1: Differences in stereocilia bundle heights and fluorescence signals (phalloidin and anti-EPS8, WT vs. *Eps8* KO).

**Analysis of bundle heights and phalloidin signal across cochlear regions and hair cell types in neonatal P5 WT and *Eps8* KO mice (Fig 10A):**

To challenge our approach and demonstrate its utility, we conducted a study involving 18 cochlear stacks from neonatal P5 mice, with samples from three wild type (WT) animals and three *Eps8* KO mice, which have previously been shown to have abnormally short stereocilia [37]. Each animal's cochlea was analyzed across three regions (Base, Middle, and Apex), and within each region, we studied two cell types (IHCs and OHCs). See Fig 10A (top row) for a detailed analysis of stereocilia bundle height variations between WT and KO. The violin plots were generated using data processed through the VASCilia plugin. Consistent with prior studies, VASCilia showed a gradient of increasing bundle heights along the base-mid-apex tonotopic axis for both WT and *Eps8* KO mice, with *Eps8* KO mice showing significantly shorter bundles than WT. S1 and S2 Tables summarize bundle height distributions (mean, SD, median, N) and the pairwise tonotopic differences. They show a robust Base→Middle→Apex increase in WT (IHC/OHC) and KO OHCs, with the only non-significant step being Base vs. Middle in KO IHC (Holm-adjusted tests).

See Fig 10A (bottom row) for the results of our challenge case study on the accumulated phalloidin signal between WT and *Eps8* KO mice in three regions (Base, Middle, and Apex) for two cell types (IHC and OHC). Intensities were computed as the 3D sum within each bundle and global Min-Max normalized across all WT+KO samples to place values on a scale of 0 to 1. In this analysis, we applied mode 2, discussed in Sect 3, separately to each dataset. For a given data set $D$, we set $n$ to the largest depth shared by all bundles within $D$; equivalently, $n = \min\{\text{bundle depths in } D\}$. This dataset-specific choice of $n$ ensures that within each data set, all bundles contribute the same number of layers. The IHC WT groups showed the highest normalized intensities in the Base region (median 0.84), with a tonotopic gradient of decreasing intensity from Base to Apex across all groups. KO groups exhibited a significant reduction in intensity compared to WT, with the OHC KO Apex group showing the lowest median intensity of 0.11. See S3 and S4 Tables, which report descriptive statistics (mean, SD, median, $N$) and pairwise contrasts (Δ, 95% CI, Hedges' $g$, Holm-adjusted $p$-values); significant comparisons are indicated.

**Analysis of anti-EPS8 signal in mid-cochlear inner hair cells of mature P30 WT and *Eps8* KO mice (Fig 10B):**

Quantitative analysis of stereocilia bundle morphology and anti-EPS8 signal in postnatal day 30 (P30) inner hair cells (IHCs) revealed pronounced structural and molecular differences between wild type (WT) and *Eps8* knockout (KO) mice. WT bundles ($n = 14$) exhibited a mean length of $3.59 \pm 0.29$ $\mu$m (SD; range: 3.12–4.28 $\mu$m), whereas *Eps8* KO bundles ($n = 15$) were significantly shorter, measuring $1.22 \pm 0.18$ $\mu$m (SD; range: 0.90–1.53 $\mu$m; Welch's $t = 25.89$, $df = 21.29$, $p = 1.4 \times 10^{-17}$). To evaluate bundle-specific anti-EPS8 labeling intensity, total fluorescence intensity was quantified within the same segmented IHC bundles. WT bundles exhibited a mean total intensity of $0.616 \pm 0.059$ (SD), while *Eps8* KO bundles showed markedly lower values of $0.094 \pm 0.012$. Normalization was performed by dividing each bundle's total intensity by the maximum total intensity observed across both WT and KO groups. This reduction was highly significant (Welch's $t = 8.63$, $df = 14.16$, $p = 5.1 \times 10^{-7}$). These findings demonstrate that the loss of *Eps8* leads to a substantial decrease in both the length of the stereocilia and the anti-Eps8 signal, consistent with its essential role in the development and maintenance of the stereocilia F-actin architecture.

### 2.4.4 Case Study 2: Fluorescence-based bundle texture analysis and classification in *Cdh23*$^{-/-}$ and *Cdh23*$^{+/-}$ hair cells.

**Quantifying bundle disorganization (*Cdh23*$^{-/-}$ vs. *Cdh23*$^{+/-}$) (Fig 12A):** Here we present an illustrative case study demonstrating VASCilia's utility beyond bundle-height and intensity measurements. We analyzed 3D image stacks from *Cdh23*$^{v-6J/v-6J}$ mice, referred here as *Cdh23*$^{-/-}$ [39] to quantify texture differences (Gray-level Co-occurrence Matrix (GLCM) and Local Binary Pattern (LBP)) between genotypes. We compared heterozygous controls (*Cdh23*$^{+/-}$; $n = 187$ crops) with homozygous knockouts (*Cdh23*$^{-/-}$; $n = 176$ crops). *Cdh23*$^{+/-}$ bundles were well organized, whereas *Cdh23*$^{-/-}$ bundles were visibly disorganized, consistent with the expected phenotype.

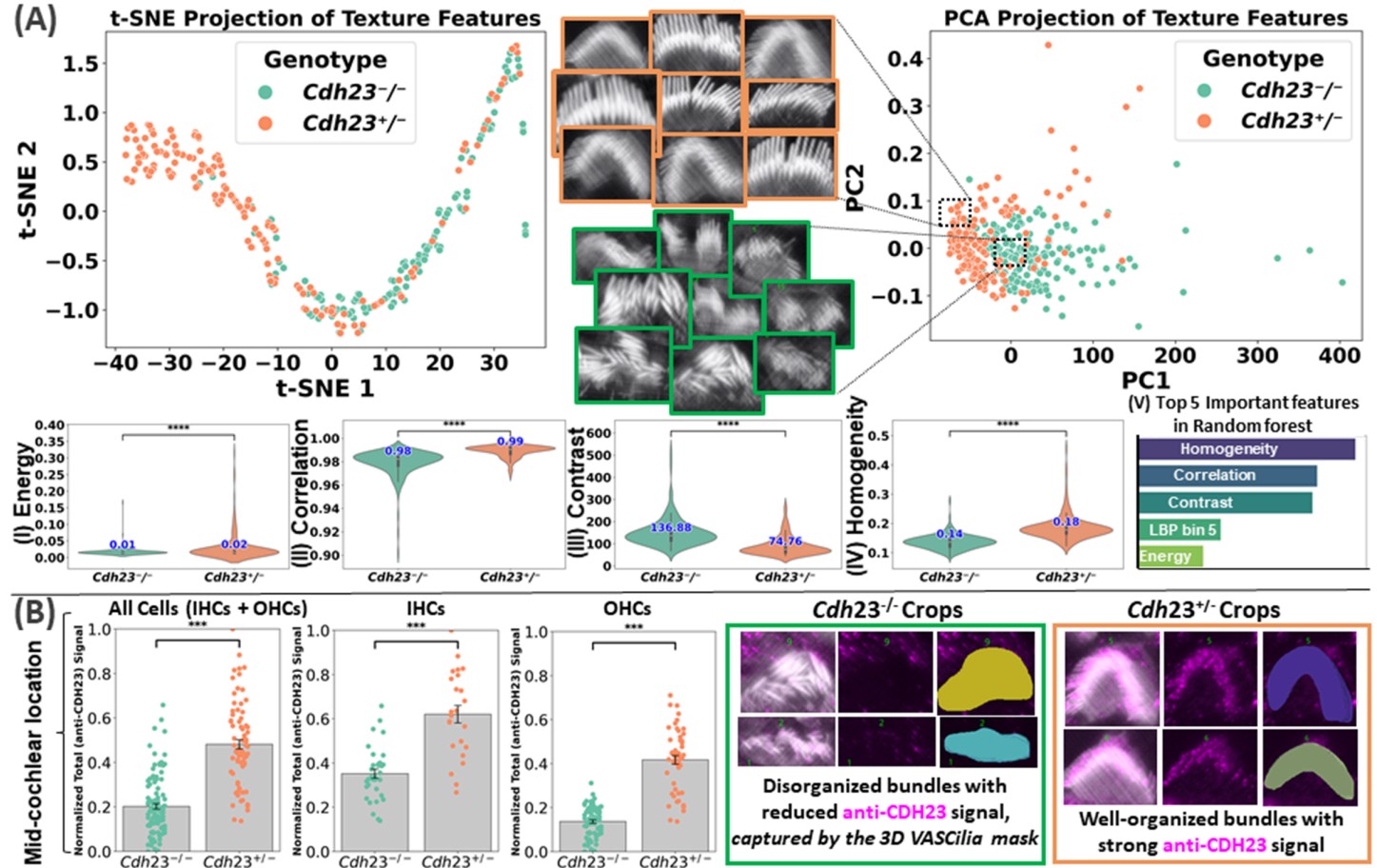

**Fig 12.** **This figure presents the texture analysis displayed at the top as panel (A), and the anti-CDH23 fluorescence quantification analysis included at the bottom as panel (B).** Source data are provided in *Supplementary Information S1 Data.xlsx*. (Panel A) [Top row]: t-SNE of texture features [I to IV] and PCA projection show clear genotype separation in feature space, with crops from the two genotypes providing visual representation. (Panel A) [I to V]: GLCM energy by genotype (Cdh23$^{-/-}$ bundles have lower energy than Cdh23$^{+/-}$, reflecting less uniform, more complex textures; differences are statistically significant); GLCM correlation by genotype (Cdh23$^{-/-}$ bundles exhibit lower correlation than Cdh23$^{+/-}$, indicating weaker spatial regularity; differences are statistically significant); GLCM contrast by genotype (Cdh23$^{-/-}$ bundles show higher contrast than Cdh23$^{+/-}$, consistent with greater local intensity variation; differences are statistically significant); GLCM homogeneity by genotype (Cdh23$^{-/-}$ bundles show reduced homogeneity relative to Cdh23$^{+/-}$, consistent with more uneven local texture; differences are statistically significant); and Random forest feature importance highlights key GLCM/LBP drivers. (Panel B): Anti-CDH23 immunofluorescence quantification analysis for P7 IHC and OHC stereocilia bundles from a mid-cochlear location. Each data point represents a value obtained from an individual bundle, normalized to the maximum value within the dataset. Exemplar images on the right show stereocilia bundles stained with phalloidin (white) and anti-CDH23 antibody (magenta), with the automatically generated stereocilia bundle mask overlaid.

To quantify these differences, we extracted texture features from segmented bundle crops using VASCilia. Specifically, we computed GLCM features—contrast, correlation, energy, and homogeneity—and LBP [40] histograms for each crop. Cdh23$^{-/-}$ samples had higher median contrast (136.88 vs. 74.76), consistent with more heterogeneous texture; lower correlation (0.98 vs. 0.99), reflecting weaker spatial regularity; lower energy (0.0147 vs. 0.0157), indicating less uniformity; and lower homogeneity (0.1368 vs. 0.1782), consistent with greater local contrast. Two-sided Mann–Whitney U tests showed highly significant differences across all GLCM features ($p < 0.0001$). Several LBP bins (0, 1, 2, 5, 6, 7, 9) also differed between genotypes ($p < 0.01$), reflecting shifts in local texture patterns. Dimensionality reduction (PCA and t-SNE) visually revealed a clear separation between genotypes in feature space. To quantitatively validate this observation, we

trained a random forest on the extracted features (training $n = 290$; test $n = 73$), which achieved 94.5% accuracy on held-out crops (4/73 misclassified). The most important features aligned with those that were statistically significant (Fig 12A); representative examples are shown in S5 Fig.

**Anti-CDH23 fluorescence quantification analysis ($Cdh23^{-/-}$ vs. $Cdh23^{+/-}$ (Fig 12B):** To examine how loss of cadherin-23 affects anti-CDH23 staining, we quantified total fluorescence intensity in IHCs and OHCs bundles from $Cdh23^{-/-}$ and $Cdh23^{+/-}$ mice. Data were obtained from 7 $Cdh23^{-/-}$ and 6 $Cdh23^{+/-}$ datasets, all from the mid-cochlear location. Values were min–max scaled using the global all cells range across both genotypes and cell types. As shown in Fig 12B, across all bundles, $Cdh23^{-/-}$ cells ($n = 112$) measured $0.202 \pm 0.012$ SEM, whereas $Cdh23^{+/-}$ cells ($n = 76$) had a mean normalized intensity of $0.482 \pm 0.021$ SEM. By cell type, IHC bundles were highest: $Cdh23^{-/-}$ ($n = 34$) $0.352 \pm 0.021$ SEM vs. $Cdh23^{+/-}$ ($n = 24$) $0.621 \pm 0.040$ SEM. OHC bundles showed the same trend: $Cdh23^{-/-}$ ($n = 78$) $0.137 \pm 0.007$ SEM vs. $Cdh23^{+/-}$ ($n = 52$) $0.417 \pm 0.019$ SEM. These results demonstrate a robust, genotype-dependent reduction in anti-CDH23 bundle fluorescence in $Cdh23^{-/-}$ hair cells across both IHCs and OHCs, consistent with cadherin-23 being required to maintain normal stereocilia organization and F-actin integrity in the mid-cochlea.

**2.4.5 Tonotopic position prediction (BASE, MIDDLE, APEX).** One of the fascinating features of cochlear hair cell bundles is the so-called "tonotopic" organization, wherein the bundle heights follow a pattern of increasing lengths as a function of position along the length of the cochlea, see Fig 1. Since the frequency of sound detected follows a similar pattern along the length of the cochlea, we were motivated to generate a model that can accurately assess the tonotopic position of the imaged region of the cochlea. To conduct this task, we trained a classification CNN. For more details about the training and architecture, see Sect 4.8. This method has demonstrated robust performance, achieving a subject-based accuracy of 97%; 28 out of 29 stacks were correctly identified. The single observed misclassification in our testing data involved a stack from the MIDDLE being incorrectly predicted as BASE, which we speculate was due to the close proximity and overlapping characteristics of these two cochlear regions. The covariance matrix for the regions BASE, MIDDLE, and APEX is in Fig 13.

To gain insights into the focus areas of our model during prediction, we employed Gradient-weighted Class Activation Mapping (Grad-CAM), a powerful visualization tool used to explain deep learning-based predictions. Grad-CAM helps visually identify which parts of an image are pivotal for a model's decision-making process. Grad-CAM works by capturing the gradients of the target label, which is the correct class in this scenario, as they propagate back to the final convolutional layer just before the softmax. It then uses these gradients to weigh the convolutional feature maps from this layer, generating a heatmap that visually emphasizes the image regions most critical for predicting the class label. Grad-CAM visualization is shown in Fig 13.

**2.4.6 High-throughput computation of bundle orientation.** Understanding the precise orientation of stereocilia bundles in the cochlea is critical for deciphering the intricate mechanics of hearing. These hair-like structures are arranged in a "V" shape due to a phenomenon called planar cell polarity (PCP). PCP ensures that neighboring hair cells and their

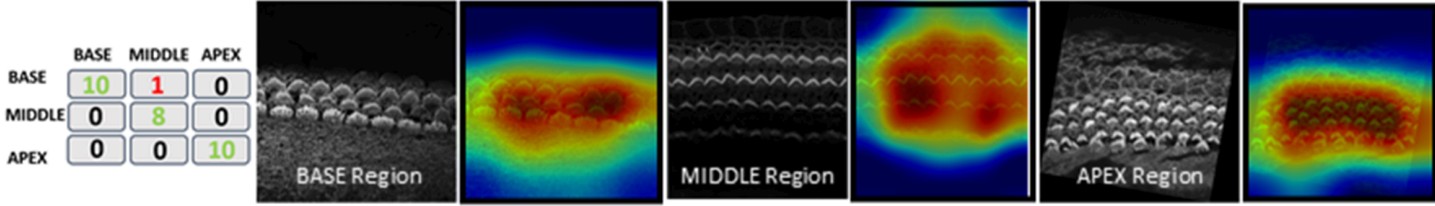

**Fig 13. left: covariance matrix for model prediction, others: Grad-CAM Visualization for Base, Middle, and Apex; Resized and Overlaid Response from the Last Convolutional Layer Highlighting Focus on Bundles During Decision Making.** Source data are provided in *Supplementary Information S1 Data.xlsx*.

stereocilia bundles are aligned in a specific direction which plays an essential role in orchestrating the precise orientation of stereocilia bundles, which is fundamental for both the directional sensitivity and efficient mechanotransduction necessary for our sense of hearing. Studying and potentially manipulating PCP holds immense potential for developing better hearing aids, diagnosing hearing loss with greater accuracy, and understanding the mechanisms behind various auditory disorders. In addition, it provides information on the effects of genetic mutations and environmental factors on hearing and facilitates comparative studies that explore evolutionary adaptations in hearing.

We developed two automated mechanisms, "Height only" and "Height and Distance", in our plugin to automatically obtain the orientation of the stereocilia bundles based on our 3D segmentation masks:

For the 'Height only' approach, our method pinpoints and records the lowest points on both the left and right sides of the centroid within the 2D projection. Conversely, the 'Height and Distance' method not only locates these points but also measures distances from a specific peak point to identify the most remote and lowest points on each side. While the 'Height only' approach functions perfectly with cells that exhibit a V-shape structure as clear in Fig 14, it is less effective for cells with atypical shapes, commonly encountered in the apex region. This limitation led to the development of the 'Height and Distance' method, which reliably accommodates cells with irregular shapes, see Fig 15.

Following the orientation calculations, the script proceeds to generate lines and angles for each region. It connects the corresponding left and right points and calculates angles using the arctan2 function for deltaY and deltaX, providing a precise angular measurement. These orientation points, lines, and computed angles are then visualized in the Napari viewer through distinct layers, specifically designated for points, lines, and text annotations. This visualization is enhanced with carefully chosen colors and properties to ensure clarity and optimal visibility. To ensure dynamic interactivity within our visualization tool, event listeners are embedded to reflect any adjustments in orientation points directly on the orientation lines and measurements.

For Fig 14, the Fiji measurements had a mean of 94.04°, median of 92.32°, and standard deviation of 8.83°, while VASCilia reported a mean of 93.36°, median of 92.47°, and standard deviation of 8.44°. The Wilcoxon signed-rank test gave a p-value of 0.0652, and the paired t-test yielded a p-value of 0.0834, indicating no statistically significant difference. Similarly, for Fig 15, Fiji results showed a mean of 87.94°, median of 87.73°, and standard deviation of 7.80°, while VASCilia produced a mean of 86.88°, median of 85.75°, and standard deviation of 5.90°. The Wilcoxon test returned a p-value of 0.1901, and the paired t-test gave a p-value of 0.2738, again confirming no statistically significant difference. These results validate that VASCilia produces automated angle measurements that are consistent with manual Fiji annotations. See S5 and S6 Tables that record all comparison values.

We also quantified bundle orientation angles in PCP-deficit cochleae [41] to assess generalization. The early postnatal Pcdh15-R245X mouse cochleae were acutely dissected, fixed with 4% PFA, stained with phalloidin to visualize the stereocilia bundles, then mounted and imaged with Leica SP8 confocal microscopy using a 63x1.3NA lens. Validation against manual Fiji measurements showed comparable means without statistically significant differences (S9 Table; S4 Fig).

**2.4.7 Hair cell classification (IHC, OHC1, OHC2, OHC3).**  The categorization of hair cells into IHC, OHC1, OHC2, and OHC3 in cochlear studies is not merely a morphological distinction but is deeply tied to their physiological roles, susceptibility to damage, and their distinct roles in the auditory system. Higher throughput classification will enable more nuanced research exploration, diagnostics, and treatments in audiology and related biomedical fields. Identifying these rows manually is a time-consuming and laborious process. For this reason, we have enhanced our plugin with mechanisms to identify the four rows using three strategies: KMeans, Gaussian Mixture Model (GMM), and deep learning. As a result, this can significantly automate the process, providing high-throughput results for all cells in the current stack in just one second.

1. KMeans: We have implemented KMeans clustering due to its simplicity and efficiency in partitioning data into distinct clusters based on their features. Specifically, we use KMeans to categorize hair cells into four clusters, leveraging

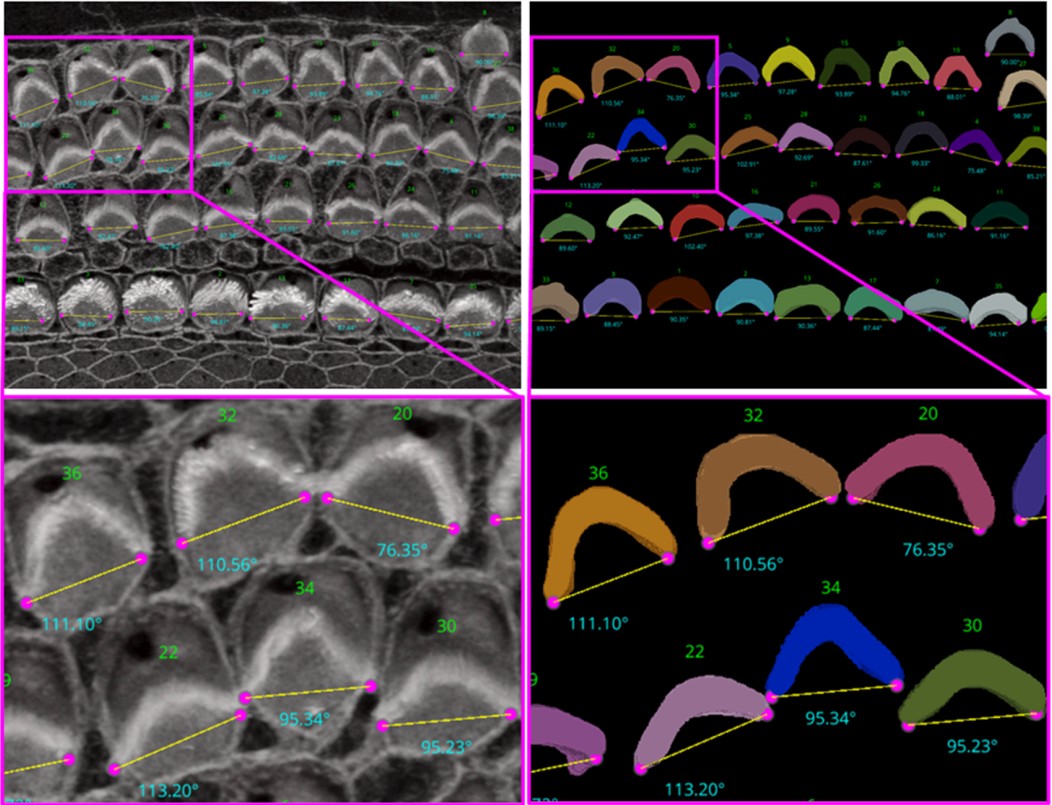

**Fig 14. Automated Computation of Stereocilia Bundle Orientation Using a Height-Only Method.** Top Left: illustrates the bundle orientations superimposed on the raw data, Top right: displays the 3D segmentation masks with bundle orientation highlighted. Bottom Right and Left are cropped regions for a closer look. Source data are provided in *Supplementary Information S1 Data.xlsx*.

attributes such as proximity and density for grouping. This method calculates the centroids of the data points and iteratively minimizes the distance between these centroids and their assigned points, making it highly suitable for fast, general clustering tasks. KMeans proves particularly effective when IHCs and OHCs are distinctly separated and organized in linear rows, especially as it performs optimally without the presence of outliers or significant overlap between rows.

2. Gaussian Mixture Model: The Gaussian Mixture Model is a probabilistic model that assumes all the data points are generated from a mixture of several Gaussian distributions with unknown parameters. By fitting the GMM to the 'y' coordinates of hair cell endpoints, we can predict the cluster for each cell, which aids in categorizing them into distinct groups based on their vertical spatial alignment. This approach is particularly useful in scenarios where the clusters might have different variances, which is often the case in biological tissues. The ability of GMM to accommodate mixed distribution models helps in accurately classifying cells into four predetermined clusters. The Gaussian Mixture Model (GMM) offers flexibility over KMeans by accommodating clusters of varying shapes, sizes, and densities, which is crucial in biological data where such variations are prevalent. Unlike KMeans, which assumes spherical clusters and hard partitions, GMM models the probability of each point's membership in potential clusters, allowing for soft clustering.

3. Deep Learning: While KMeans and GMM provide robust and instant results for many samples, see first row of Fig 16, however, these statistics-based methods can struggle with complex configurations, particularly in samples with outliers, overlapping rows, or non-linear, curvy arrangements. For these challenging scenarios shown in second and third

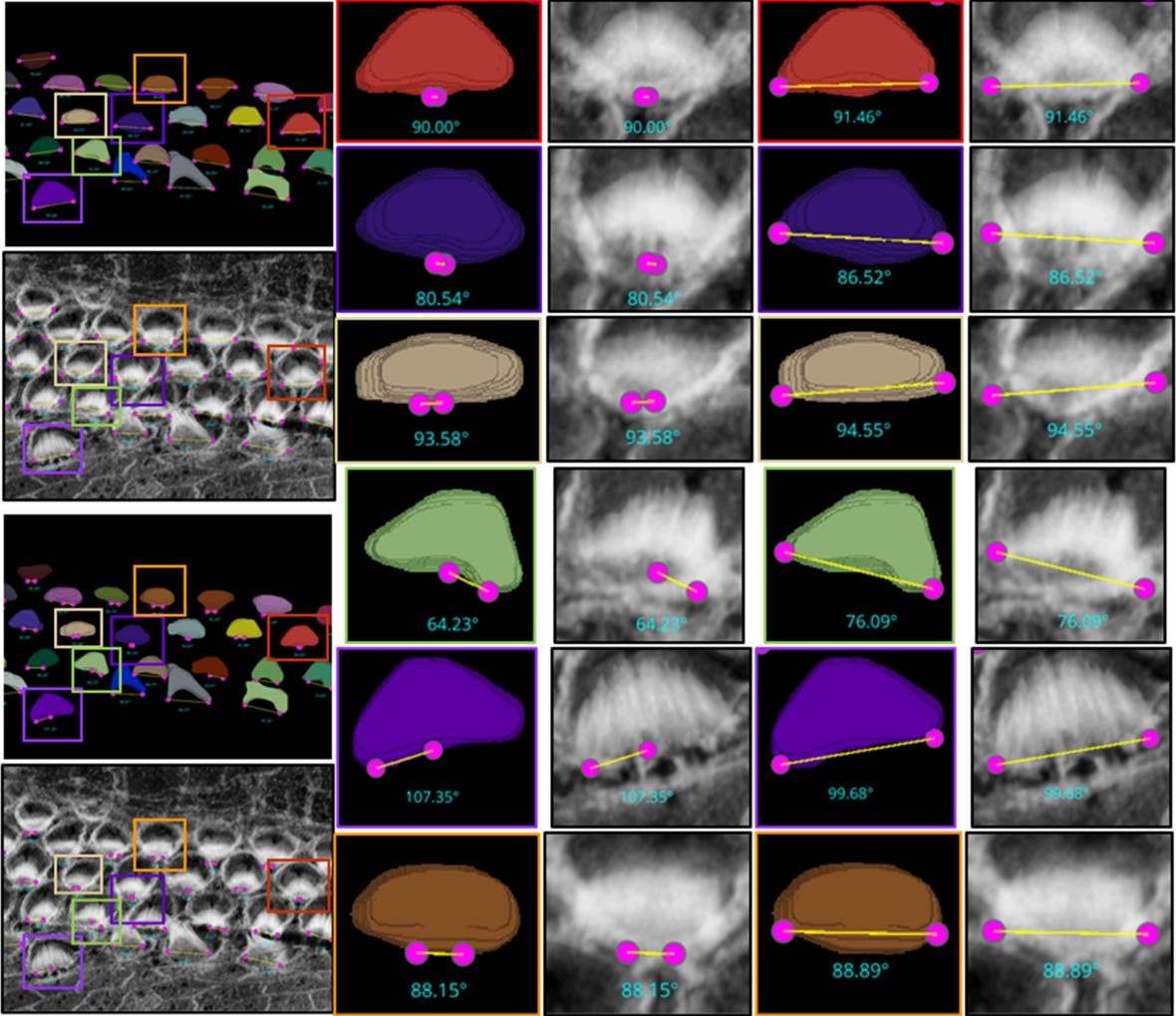

**Fig 15**. **The limitations of the Height-Only orientation computation.** While the Height-Only method excels with cells exhibiting a clear V-shape Fig 14, it struggles with some apical region hair cells that lack this distinct structure. To address these challenges, we developed the Height and Distance approach, which effectively handles a wider variety of cell shapes. The visual comparison includes the frames from which the crops are taken (first column), Height-Only results superimposed on the 3D segmented labels (second column) and the original images (third column), alongside the height and distance results superimposed on the 3D segmented labels (fourth column) and the original images (fifth column). We observe that the Height and Distance method overcomes the limitations of the Height-Only computation. Source data are provided in *Supplementary Information S1 Data.xlsx*.

row of Fig 16, we have developed a deep learning approach that utilizes multi-class classification to accurately identify and categorize all hair cell bundles, effectively handling the spatial and structural complexities inherent in such data. Check Sect 4.9 about the architecture and model training details.

For inference, we focused exclusively on stacks from the Apex region, where the cells frequently display non-linear growth, are closely packed, and often overlap, leading to higher error rates with KMeans and GMM methods. Conversely, cells in the Base and Middle regions are typically well-aligned. We conducted two experiments: In the first, we tested all Apex data, including 41 datasets from both training and testing phases. In the second experiment, we evaluated 25 stack datasets that were not included in the training set to verify the model's performance on both seen and unseen data.

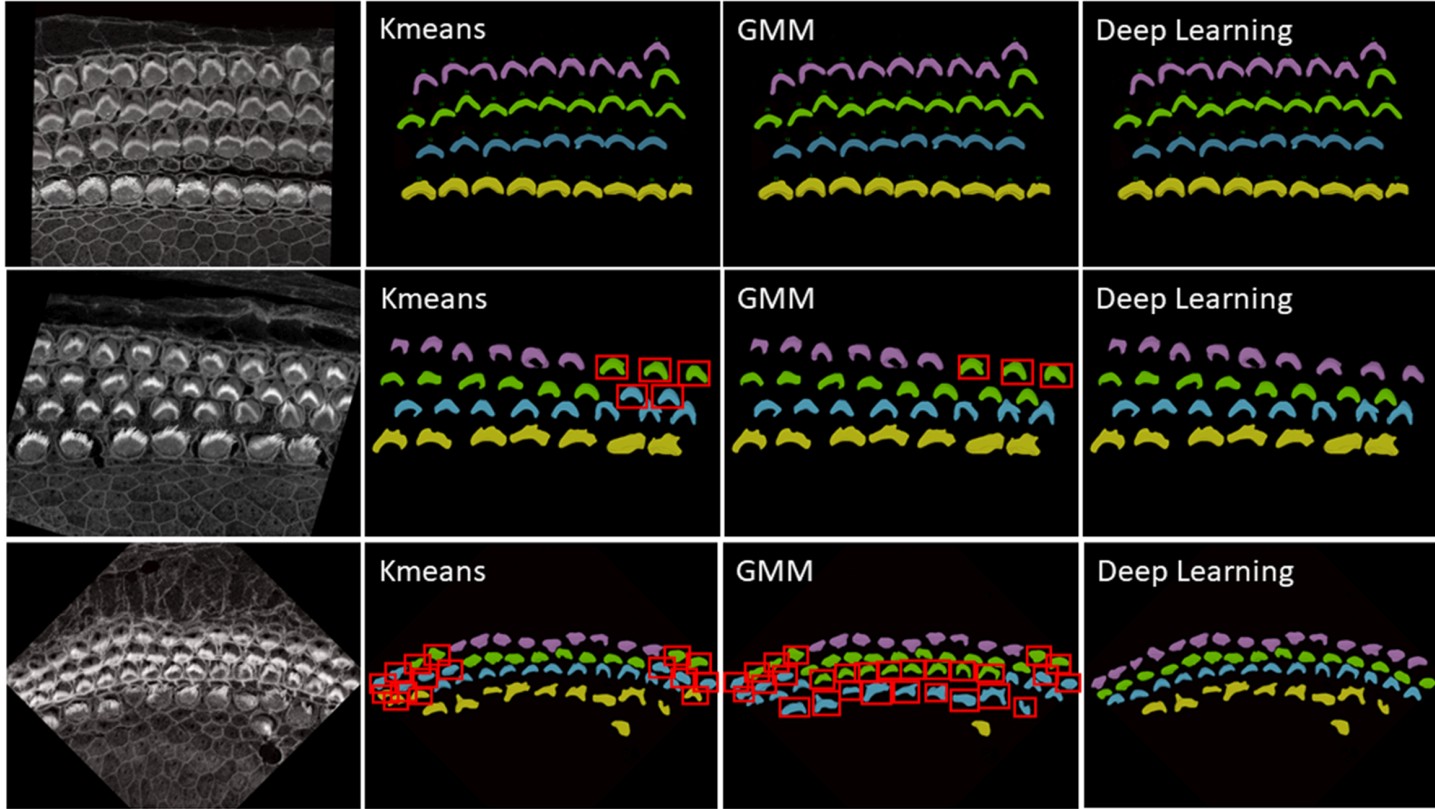

**Fig 16**. **First row: Successful cases for all methods—KMeans, GMM, and Deep Learning—in accurately clustering the four rows into their respective categories are clearly demonstrated.** This scenario represents an ideal case where each row is well-separated, linearly aligned, and free from outliers that simplify the task of accurate clustering. Second and third rows: Failure cases for KMeans and GMM in accurately clustering the four rows into their respective categories are evident: IHC1 in yellow, OHC1 in cyan, OHC2 in green, and OHC3 in magenta. These traditional methods often struggle to precisely segregate the rows due to their inherent limitations in handling complex data distributions, outliers, and overlapping clusters. In contrast, Deep Learning significantly outperforms both KMeans and GMM, providing accurate and reliable clustering for all cell types. Errors are represented by red bounding boxes. For the sample in the second row, there are Five errors in KMeans, three errors in GMM, and no errors with deep learning. For the sample in the third row, there are Sixteen errors in KMeans, twenty eight errors in GMM, and no errors with deep learning

Fig 17A and 17B illustrate the error rates by method and dataset to assess performance per sample. Fig 17C and 17D present heatmaps of the errors by method and sample, offering a more visual representation of performance with a color map transitioning from black at the bottom to yellow at the top. These figures highlight the pronounced error frequency in the KMeans and GMM methods compared to our deep learning approach. Fig 17E and 17F display the cumulative errors, providing insight into how errors accumulate linearly per dataset and method. It is notable that the errors with KMeans and GMM, represented in blue and brown, show a steep increase, whereas the deep learning method demonstrates a more consistent error rate. In Experiment 1, across a total of 1824 cells, the error counts were 202 for KMeans, 257 for GMM, and 12 for Deep Learning. In contrast, Experiment 2 involved 1102 cells, with error counts of 136 for KMeans, 169 for GMM, and 11 for Deep Learning. We also conducted a one-way ANOVA (Analysis of Variance) test. This statistical test is used to determine whether there are statistically significant differences between the means of three or more independent (unrelated) groups. We found that there is significant differences between groups with F-value equal to 10.45, A higher F-value indicates a greater degree of variation among the group and P-value equal to 0.0001 which reject the null hypothesis of the ANOVA test. The null hypothesis for an ANOVA test states that all group means are equal.

## [A] Error Rate for all Apex Data

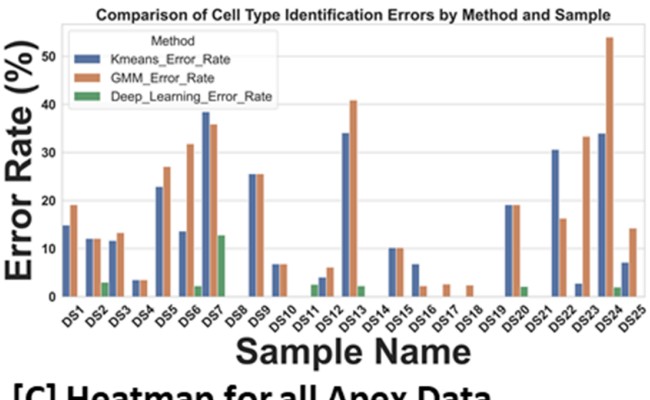

## [B] Error Rate for APEX testing set

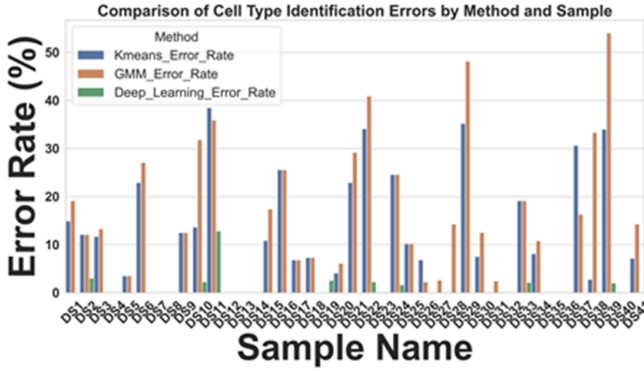

## [C] Heatmap for all Apex Data

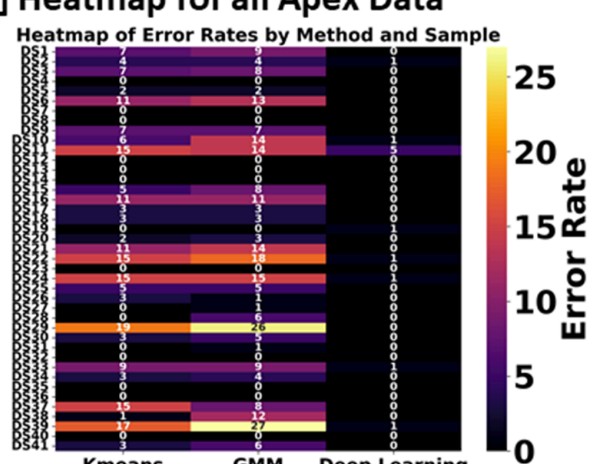

## [D] Heatmap for APEX testing set

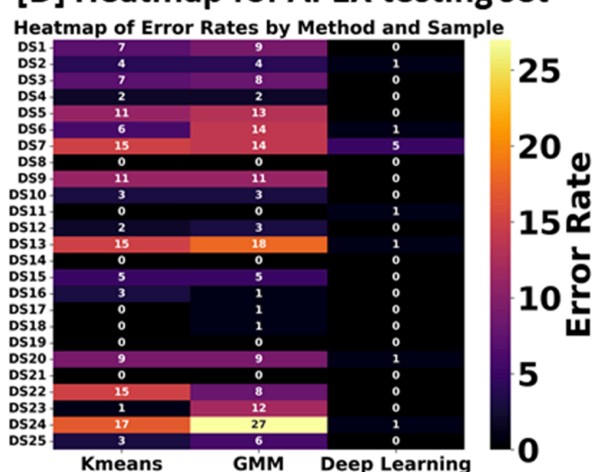

## [E] Cumulative error for all Apex Data

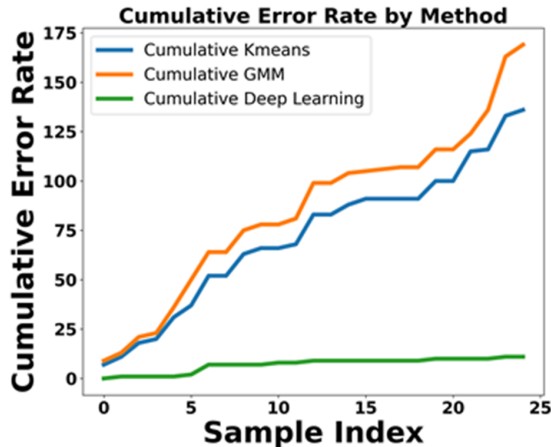

## [F] Cumulative error for only testing set

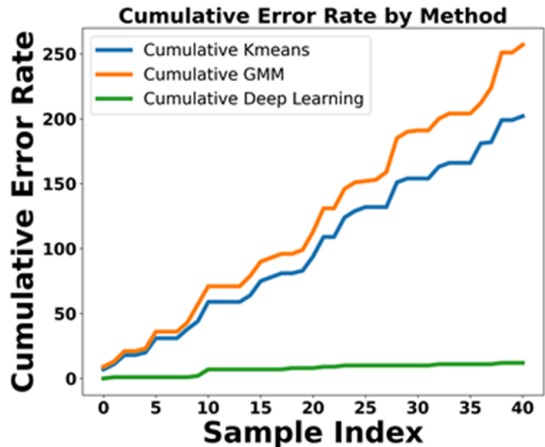

**Fig 17**. **Comprehensive Examination of Error Rates, Heatmaps, and Cumulative Errors for Cell Type Identification in the Apex Region.** Subplots (A, B, E, and F) illustrate error rate and cumulative errors, with blue representing KMeans, brown for GMM, and green denoting Deep Learning. Subplots (C and D) utilize the 'inferno' colormap to depict error rates, transitioning from black (low errors) to yellow (high errors), providing a visual gradient of error severity. This color-coded representation aids in distinguishing the methodologies applied across different datasets and highlights the specific error dynamics associated with each method.Source data are provided in *Supplementary Information S1 Data.xlsx*.

## 3 Discussion

VASCilia, displayed in Fig 18, stands out in several unique ways: 1) Purpose-Built for Inner Ear Hair Cell Bundle Research: VASCilia is the first Napari plugin and software analysis tool specifically designed for 3D segmentation of stereocilia bundles. It is equipped with comprehensive analysis capabilities and machine learning models tailored for the inner ear imaging community. 2) User-Friendly Interface: VASCilia features an intuitive, easy-to-install interface that allows users to navigate the tool with minimal effort or expertise. 3) Flexible Application and Fine-Tuning: The flexibility of VASCilia allows users to directly apply it to their own datasets, ranging from mice to humans. Additionally, users can fine-tune their own models using the tool's built-in training module. 4) Batch Processing Capability: VASCilia provides a robust batch processing feature, enabling users to process multiple samples automatically. With automated filtering and rotation of each image stack, all subsequent steps in the pipeline are streamlined. Users simply input the names and paths of their sample files, and VASCilia generates results for each sample, storing them in separate folders with all intermediate results saved for easy access. 5) Review and Collaboration: VASCilia offers a distinctive feature for reloading analysis results from a pickle file, allowing multiple users to review results at a later time, facilitating collaboration and simplifying the review process. All workflow steps and buttons are documented with step-by-step screenshots in https://ucsdmanorlab. github.io/Napari-VASCilia/overview.html.

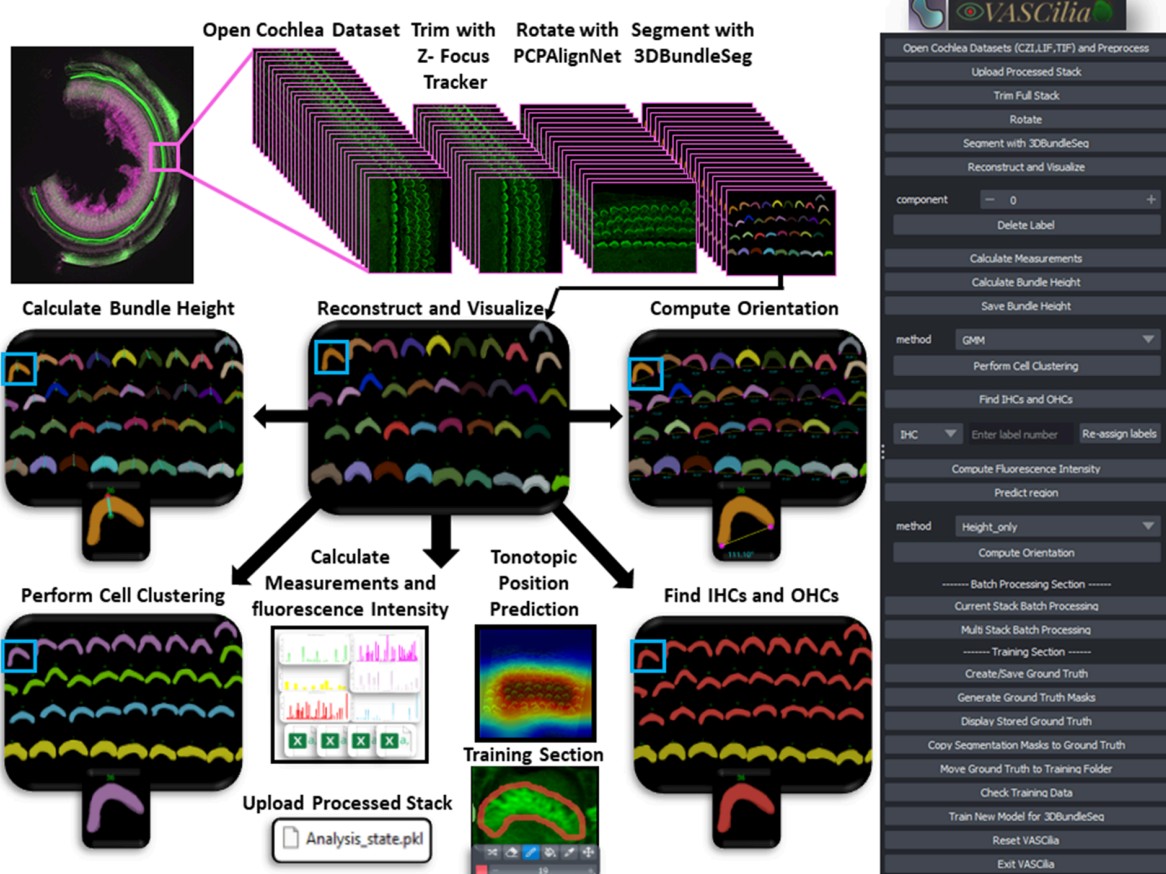

**Fig 18**. **VASCilia enables the ear research community to process their cochlea samples through an end-to-end workflow, all within a user-friendly interface.**

VASCilia delivers end-to-end 3D segmentation and quantitative analysis of stereocilia bundles; here we synthesize robustness evidence and key biological insights from each Results section.

In Sects 2.1 and 2.2, two preprocessing steps most strongly stabilize downstream analysis. First, ZFT-Net yields near switch-free PCZ→CCZ→NSZ focus labels along the $z$-axis, so segmentation and quantification operate on cellular-clarity planes rather than out-of-focus frames. Second, PCPAlignNet places each stack in a PCP-aligned reference frame, reducing rotational variance and standardizing height, intensity, orientation, and tonotopic measurements.

In Sect 2.3, accurate 3D instance segmentation of cochlear stereocilia is intrinsically challenging, due to nonspecific staining; close apposition within and across rows; noise, intra-cell signal dropout, intensity variation; and tonotopic shape heterogeneity (Fig 3). Despite these factors, our approach consistently separated closely apposed bundles and maintained detection in low-signal frames, indicating robustness to common experimental artifacts. To highlight variability in segmentation depth as a function of bundle orientation (upright vs. flat) and phalloidin-signal continuity, and to show that the segmentation excludes the cuticular plate, S1A and S1B Fig presents two bundles from a representative apical P21 cochlear stack, including all z-planes with and without overlays; orthogonal slice views (XY, XZ, YZ); and 3D volume renderings.

In Sect 2.4.1, VASCilia height measurements show human-level reliability, matching expert annotations with strong agreement, supporting its use for large quantitative studies. Beyond agreement with experts, a key methodological advantage is that our length metric is rotation-invariant because it traces bundles through the $z$-stack rather than relying on projected height. As shown in S3A–S3C Fig, rotated OHC1 bundles traverse more layers (depth = 10) than the straighter OHC2/3 bundles (depth = 5), yet the computed lengths are comparable. For example, OHC1 cell bundles with IDs 17 and 18 measure 1.65–1.69 $\mu$m and cell bundle 16 is 1.75 $\mu$m, comparable to OHC2 cell bundle 38 at 1.71 $\mu$m. The apparent visual foreshortening effect of rotated bundles reflects viewing angle, not true length; the increased z-plane depth afforded by the 3D imaging data accounts for these angle differences. In ambiguous cases, specifically when a bundle does not exhibit a typical V-shape, we performed optional manual refinement. The plugin includes event handlers that allow users to manually adjust the highest and lowest points of each bundle. These adjustments are easily made through the intuitive 3D viewer in the Napari interface, enabling researchers to refine measurements as needed.

In Sect 2.4.2, one potential limitation when measuring total intensity between bundles is the variability in segmentation depth across cells, which can arise from differences in fluorescence signal strength (phalloidin here), imaging depth, or biological heterogeneity. This variability may cause some bundles to appear shallower, potentially affecting their total intensity measurements. Although such differences tend to average out when large numbers of cells are analyzed, minimizing their impact on overall trends and group comparisons, we also provide a dedicated option for researchers who wish to control for this factor. Specifically, *VASCilia* outputs a CSV file that records per-layer intensity values for each bundle. We extended the analysis code to offer three alternative aggregation modes for per-bundle fluorescence intensity (shown in Fig 11): (i) the sum of raw intensities across all detected layers for each bundle; (ii) the sum restricted to the top $n$ layers, where $n$ is the largest depth shared by all bundles being compared; and (iii) the sum obtained by padding shallower bundles to $n$ layers by repeating the bundle's weakest-layer value, where $n_{\max} = \max_i d_i$ is the maximum depth across bundles.

To illustrate depth-controlled comparisons, Fig 11 presents four depth-matched examples—two IHCs (Cell #3 and Cell #7; yellow) and two OHCs (Cell #26 and Cell #18; red). With depth held constant, the total 3D intensity becomes primarily driven by morphology (bundle size/footprint) and F-actin density. Consistent with this, Cell #18 shows the lowest total intensity because it is the smallest and finest, whereas Cell #3 shows the highest total intensity, reflecting a larger bundle and denser phalloidin signal.

In Sect 2.4.3, which examines the relationship between bundle height and phalloidin intensity, we observed that height increases tonotopically from base to apex in both WT and *Eps8* KO, while *Eps8* loss yields consistently shorter bundles. Phalloidin intensity shows the opposite trend—highest at the base and decreasing toward the apex across groups—with KO animals reduced relative to WT (e.g., lowest in OHC KO at the apex). Although basal bundles are shorter than apical

bundles, the higher basal intensity is consistent with a higher packing density of F-actin rather than length. Prior work also reports that IHC stereocilia are thicker than OHC stereocilia [42–44]. See S11 Fig for a depth-matched visual comparison of base vs. apex in WT; the same qualitative pattern holds in KO. In summary, phalloidin-measured actin content does not necessarily increase with bundle height across WT and *Eps8* KO P5 animals.

In Sect 2.4.4, texture analysis captured the expected phenotype: homozygous $Cdh23^{-/-}$ bundles exhibited more heterogeneous and less regular stereocilia organization than heterozygous controls, consistent with visible disorganization. Across metrics, $Cdh23^{-/-}$ crops showed systematic shifts (e.g., lower energy, homogeneity, and correlation with higher local contrast), and these differences were separable in low-dimensional embeddings; a simple classifier trained on the same features generalized well to held-out data.

In Sect 2.4.5, we noticed how Grad-CAM (Fig 13) serves as a sanity check to confirm the model's reliance on accurate features for its predictions. Interestingly, Grad-CAM revealed that the model predominantly focuses on the hair cell bundles to distinguish between the BASE, MIDDLE, or APEX regions. Future directions include exploring the possibility of developing a foundation model capable of accurately predicting cochlear regions across various datasets with different staining and imaging protocols. This effort would undoubtedly require extensive data collection. We believe this work demonstrates that this task is feasible. The Napari-based graphical user interface and interactive visualization tools presented here should help make these methodologies and trained models accessible to the average researcher. We hope these results help motivate collaborative efforts to gather and share more comprehensive imaging data within the inner ear hair cell research community.

In Sect 2.4.6, Bundle orientation is biologically meaningful because planar cell polarity (PCP) aligns stereocilia and underlies directional sensitivity and mechanotransduction. Quantifying orientation therefore provides an objective readout of PCP integrity. The time-consuming nature of the manual PCP measurement methods highlights the need for more automated and efficient techniques to further advance our understanding of this critical process. We identified an automated Fiji plugin, PCP Auto Count, for quantifying planar cell polarity and cell counting [45]. This plugin relies on identifying "chunks" (the hair cell apical surface) and "caves" (the fonticulus) to measure the orientation between their centers. However, this approach was not effective with our images due to the absence of clear caves, especially in low-contrast images. The reliance on distinct caves for orientation measurement poses a limitation for datasets where such features are not consistently visible or distinguishable. Automated angles closely matched manual Fiji measurements on both our datasets and an independent PCP-deficit dataset [41] , indicating human-level reliability. See S5, S6, and S9 Tables that record all comparison values.

In Sect 2.4.7, automated row identification (IHC, OHC1–3) proved reliable even in challenging apical stacks where rows curve and overlap. Compared with unsupervised clustering (KMeans/GMM), the learned classifier handled outliers and nonlinear row geometry substantially better, reducing row-assignment ambiguity in the apex. In Fig 17, errors for KMeans and GMM (blue and brown) increase steeply, whereas the deep-learning model shows a more stable error rate.

As demonstrated in the case studies and generalizable beyond them, VASCilia enables high throughput, standardized quantification. Once 3D instance segmentation is available, the pipeline automatically extracts height, depth-matched phalloidin intensity, orientation, row identity, tonotopic position, 2D and 3D morphology metrics, and texture features across thousands of bundles. This reduces analyses that would typically require months of manual curation to hours, yielding reproducible, statistically powered findings.

***Comparison with commercial solutions:*** While commercial software packages such as Imaris and Arivis offer advanced 3D visualization and generic segmentation capabilities, these platforms are proprietary, costly, and often operate as "black boxes" with limited transparency regarding model architecture or training data. Furthermore, they are not optimized for the specific challenges of cochlear hair bundles, such as the dense, anisotropic packing of stereocilia and non-specific phalloidin staining of all actin filaments in whole mount tissues. In contrast, VASCilia is fully open-source and specifically tailored for hair-cell research. It allows users to transparently inspect and fine-tune segmentation models,

ensuring reproducibility and providing a standardized, high-throughput workflow that fills a critical gap not addressed by general-purpose commercial tools.

***Limitations and future work:*** All downstream features in VASCilia are only as accurate as the underlying 3D instance segmentation. While the model generalized across multiple datasets analyzed here, large distribution shifts (e.g., vestibular hair cells, different stains/microscopes), strong preprocessing changes (e.g., deconvolution/denoising), or very low SNR can degrade segmentation quality and thus bias derived measurements. In such cases, domain adaptation via fine-tuning on a small, labeled subset of the target dataset is advisable. We maintain and continuously update the segmentation models and training recipes in our public repository to facilitate such adaptation.

Extending segmentation from bundle level to *single stereocilium* resolution would enable higher-detail analyses, including per-bundle stereocilia counts and row-specific length distributions across the three rows within each bundle. Such capabilities would strengthen quantitative comparisons across conditions but will likely require much higher-resolution, higher-SNR datasets (e.g., STED or expansion microscopy) and specialized training data with reliable instance-level labels. Developing such a model is beyond the scope of the present work and represents an important direction for future studies.

## 4 Methods

### 4.1 Description of our 3D microscopy datasets

C57BL/6J mouse P5 and P21 cochleae were harvested and post-fixed overnight at 4°C in 4% PFA. After fixation, cochleae were dissected and tissues were permeabilized in PBS containing 0.3% Triton-X (PBST) for 30 minutes at room temperature. Alexa Fluor 568-conjugated phalloidin was applied in 0.03% PBST containing 3% NGS and incubated for 30 minutes at 23°C. Samples were washed three times with 0.03% PBST for 10 minutes each, mounted in ProLong Glass Antifade (ThermoFisher Scientific, Cat#P36980, Carlsbad, CA, USA) with a #1.5 coverslip, and imaged with a 2.5 $\mu$W 561 nm laser and a 63x 1.4NA DIC M27 objective on an 880 Airyscan confocal microscope with a 43 x 43 nm xy pixel size, 0.110 nm z-step size, and a pixel dwell time of 2.3 $\mu$s per pixel then processed with default Airyscan processing settings. All experiments were conducted under strict accordance with the recommendations and approval of the Institutional Animal Care and Use Committee (IACUC) at the University of California, San Diego (UCSD) (protocol number S23058).

### 4.2 Testing other lab datasets

**4.2.1 Lab A's datasets.**  We tested our software without any fine-tuning of the training model on another lab's (Lab A) datasets [46–48], which included samples with varying mouse ages, illumination conditions, cell counts per sample, image dimensions, and pixel sizes. The software successfully performed 3D segmentation and all subsequent analysis steps on these diverse datasets. See S6 and S7 Figs.

For this dataset, all procedures were conducted in compliance with ethical regulations approved by the Institutional Animal Care and Use Committee of Mass Eye and Ear, and in agreement with ARRIVE guidelines. Cochleae were dissected in L-15 medium, fixed in 4% formaldehyde (EMS, #15713) in HBSS for 1 hour, and washed with Ca$^{2+}$, Mg$^{2+}$-free HBSS. For mice older than P6, fixed cochleae were decalcified in 0.12 M EDTA (pH = 7.0) for 2 days, washed, then micro-dissected, and permeabilized in 0.5% Triton X-100 (Thermo Scientific, #85111) in Ca$^{2+}$, Mg$^{2+}$-free HBSS for 30 minutes. Samples were blocked with 10% goat serum (Jackson ImmunoResearch, #005-000-121) in 0.5% Triton X-100 in Ca$^{2+}$, Mg$^{2+}$-free HBSS for 1 hour, and incubated in a 1:200 dilution of rabbit anti-PKHD1L1 (Novus Bio #NBP2-13765) overnight at 4°C. Samples were washed with Ca$^{2+}$, Mg$^{2+}$-free HBSS and stained for 2 hours in a 1:500 dilution of Goat anti-Rabbit CF568 (Biotium #20099) and phalloidin CF488 (1:20, Biotium #00042) in the blocking solution. Following washes, samples were mounted on slides with the ProLong Diamond Antifade Kit (ThermoFisher Scientific #P36961) and imaged on a Leica SP8 confocal microscope (63x, 1.3 NA objective lens), or an upright Olympus FluoView FV1000 confocal laser scanning microscope (60x, 1.42 NA objective lens).

**4.2.2 Lab B's lab datasets.** We also tested the software on yet another lab's (Lab B) datasets, See S8 and S9 Figs. C57BL/6J mice were exposed at 5 weeks of age to noise at 118 dB SPL for 2 hrs, then intravascularly perfused at 8 weeks with 4% PFA. Cochleas were extracted and decalcified in 0.12M EDTA, cryoprotected in 30% sucrose, and stored for 2-3 months at -80°C. After thawing, they were dissected and blocked for one hour at room temperature in PBS with 5% normal horse serum and 0.3% Triton X-100. The tissue was then incubated overnight at 37°C with rabbit anti-ESPN (Sigma #HPA028674 @ 1:100), to label stereocilia, in PBS with 1% normal horse serum and 0.3% Triton X-100. Primary incubation was followed the next day by two sequential 60-minute incubations in an anti-rabbit secondary coupled Alexafluor 647 in PBS with 1% normal horse serum and 0.3% Triton X-100. After immunostaining, pieces were slide-mounted in Vectashield, coverslipped, and imaged on a Leica SP8 confocal with a 63x glycerol-immersion objective (N.A. = 1.3) at 38 nm per pixel in x and y and 250 nm in z.

In addition, S10 Fig shows VASCilia results on human data, despite the fact that no human data stacks were included in our training set. The stacks were imaged on a Leica SP8 confocal using a 63x glycerol immersion objective (Planapo NA 1.3) and the Lightning deconvolution package. The pixel resolution was 0.038 microns in x and y and 0.217 microns in z.

## 4.3 Z-Focus tracker for cochlear image data preparation and architecture

Our 3D image stacks consist of numerous frames, which we categorize into three distinct zones to better describe the progression of image quality and content throughout the stack:

**Pre-Cellular Zone (PCZ)**: This refers to the early frames where no cellular structures are visible. These frames likely correspond to regions outside the tissue boundaries or the initial imaging volume that has not yet captured the cellular regions.

**Cellular Clarity Zone (CCZ)**: This middle portion of the stack contains well-resolved, clearly visible cells, representing the optimal imaging conditions with a high signal-to-noise ratio. Here, the microscope achieves the clearest visualization of stereocilia bundles.

**Noise Saturation Zone (NSZ)**: The later frames where image quality degrades and noise increases significantly, likely due to reduced laser penetration, light scattering, and other optical limitations, leading to fading and distortion of cellular structures.

We define these three zones as distinct classes for training a classifier. The objective is for the classifier to automatically exclude both the **Pre-Cellular Zone (PCZ)** and the **Noise Saturation Zone (NSZ)**, retaining only the **Cellular Clarity Zone (CCZ)**. This approach will significantly reduce processing time while enabling the segmentation model to concentrate on regions containing clear cellular structures.

We have compiled a dataset of 135 3D image stacks of P5 mouse cochlea, using 125 stacks for training and validation, and reserving 10 stacks for testing. Each frame was manually annotated into one of the three defined classes, resulting in 3,355 frames for the PCZ, 1,111 for the NSZ, and 2,088 for the CCZ. After applying data augmentation through rotation, we obtained a balanced dataset with 6,710 frames for each class.

To further enhance the generalizability of the model, we incorporated 13 additional stacks from P21 mice and 22 stacks from Liberman Lab (Lab B's datasets). This expanded the training set to a total of 7,722 frames for each class, improving the model's ability to generalize across different ages, species, and microscopes.

We resized our images to 256 × 256 and tested several networks, including ResNet10, ResNet18, DenseNet121, and EfficientNet. However, our custom network, Z-Focus Tracker Net (ZFT-Net), produced the best results. ZFT-Net consists of five blocks, each containing a convolutional layer, batch normalization, ReLU activation, and a pooling layer. This is followed by an additional ReLU activation and dropout layer, and finally, two fully connected layers.

## 4.4 PCPAlignNet data preparation and architecture

Our P5 and P21 mouse datasets consist of 3D stacks with different orientations ranging from 0 to 360 due to variations in how the cochlear tissue is handled or sliced during the sample preparation process. These small discrepancies in positioning can lead to notable shifts in orientation during imaging. Additionally, the cochlea's spiral and complex three-dimensional structure contributes to the variability in the orientation of the stacks.

This variability in orientation presents a significant challenge during analysis, as the angle must be corrected so that the stereocilia bundle rows align horizontally with respect to the tissue's planar polarity axis. This is not an easy task because both the images and the bundles exhibit high variability. Even when we manually rotate the images with great precision to appear horizontal to the human eye, some rows may still differ in orientation from others, and bundles can also vary among themselves. As a result, making manual decisions to establish ground truth is extremely challenging. Therefore, we opted to define 72 classes instead of 360. We selected 70% from the stacks of the P5 and P21 data set for training, 20% for validation, leaving 10% for testing. However, these stacks are insufficient to train a robust network capable of predicting across 72 classes, as we do not actually have all the possible angles represented. To address this, we augmented our data set by rotating each frame in every stack to all 72 possible classes (i.e. ranging from 0 to 355 degrees, with 5-degree increments), see Fig 20. For training, we experimented with several models, including ResNet50, ResNet18, MobileNet, and DenseNet121. We trained each model for 50 epochs, utilizing early stopping with a patience of 3. The Adam optimizer, along with a cross-entropy loss function, was employed to optimize the model.

A key challenge that caught our attention was the need to avoid the incorporation of empty pixel values in padded regions during image rotation. To address this, we applied a rotation correction in Python, followed by cropping the largest area that excluded the empty regions. Since each frame begins at a different orientation prior to augmentation, the corrected images, after rotation and removal of empty regions, vary in scale. These variations are often beneficial for CNN training, as they improve the network's ability to generalize across different spatial representations of data. See Figs 19 and 20.

## 4.5 3D manual annotation and dataset partitioning for segmentation task

Training a 3D supervised model that efficiently segments each stereocilia bundle requires manual 3D annotation for many bundles, a process that is both cumbersome and slow. We utilized the Computer Vision Annotation Tool (CVAT) [36] to

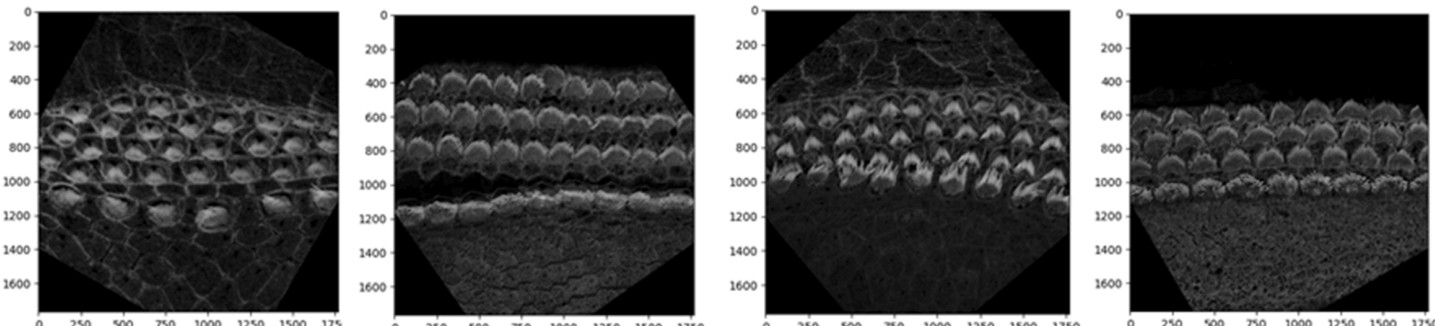

**Fig 19**. **Determining the precise orientation of the frame is challenging, as not all stereocilia bundle rows share the same horizontal alignment.** Achieving an exact degree out of 360 is practically impossible, even manually. To address this, we decided to use 72-degree intervals with 5-degree increments. We have adapted the augmentation process, shown in Fig 20, to increase the number of image samples.

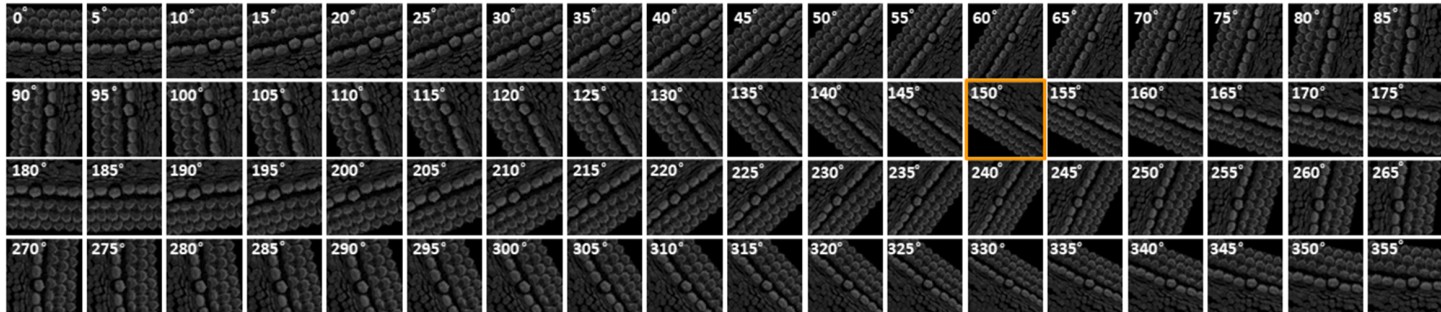

**Fig 20**. **To expand our dataset and represent all possible angles, we augmented each frame by rotating it and cropping the largest area without empty pixels from padded regions.** This process results in frames with varying perspectives and scales. In the example shown, the original slide is highlighted in **orange**, while all others are augmented versions.

annotate our 3D samples. CVAT facilitates the drawing of manual polygons with an effective tracking feature, which annotators can use to accelerate the annotation process. We manually annotated 45 stacks using the CVAT cloud application [36], assigning each 3D bundle a unique ID for precise identification. The annotated data were thoroughly inspected and refined by both the author and the biologists responsible for imaging the data.

To maintain the integrity of the data split, we divided the dataset into training, testing, and validation sets at the stack level, thus preventing the mingling of frames from different stacks during partitioning. The training set comprises 30 stacks, the validation set includes 5 stacks, and the testing set consists of 10 stacks, which are further classified into 6 typical complexity cases and 4 complex cases. The details on how many training, validation, and testing 3D instances exist in this dataset can be found in Table 5.

### 4.6 3D segmentation of stereocilia bundles using 2D detection and multi-object assignment algorithm

Our approach to the 3D segmentation task involves applying 2D detection to each frame, followed by the reconstruction of the 3D object using a multi-object assignment algorithm. We employ the Detectron2 library from Facebook Research [49], using the Mask R-CNN architecture [50] combined with a ResNet50 backbone [51] and a Feature Pyramid Network [52]. This setup leverages transfer learning from a model trained on the COCO dataset [53]. The algorithm is executed over 50,000 iterations with a learning rate of 0.00025. We focus on a single class, specifically the stereocilia bundle, with a head threshold score set at 0.5.

After getting all the 2D frame segmentation masks across all stacks, the multi-object assignment algorithm involve these steps, see Fig 4:

1. **Initialization:**
   - Set `frame_count` to zero, marking the start of the frame sequence.
   - Create an empty list `tracks` to maintain records of active object tracks.

**Table 5**. **Summary of 3D stereocilia bundle instances across different data sets.**

| Data Set | IHC | OHC | Total |
|---|---|---|---|
| Training (30 stacks) | 270 | 940 | 1210 |
| Validation (5 stacks) | 47 | 170 | 217 |
| Inference (10 stacks) | 93 | 350 | 443 |
| **Total (45 stacks)** | 410 | 1460 | 1870 |

- Load the initial frame of the stack to start the tracking process.
2. **Processing the First Frame:**
   - Detect all objects within the first frame and assign each a unique track ID.
   - Store the position and ID of each detected object in `tracks`.
   - Increment `frame_count`.
3. **Tracking in Subsequent Frames:**
   - Load the next frame to continue tracking.
   - Detect all visible objects in the current frame.
   - For each detected object:
     – Calculate the overlap area with each object in `tracks`.
     – Determine the appropriate track based on overlap:
       * If no significant overlap is found, initialize a new track.
       * If an overlap exists, assign the object to the track with the largest overlap and update the track's position.
     – Handle ambiguous cases:
       * If multiple objects overlap significantly with a single track, choose the object with the largest overlap for the track.
       * Consider initiating new tracks for other overlapping objects.
   - Increment `frame_count`.
4. **Loop Through Remaining Frames:**
   - Repeat the process in Step 3 for each new frame until the end of the stack sequence.
5. **Finalization:**
   - Assemble and output all completed tracks for further analysis.
   - Conclude the tracking algorithm. At this stage, each cell or bundle is assigned a unique ID based on the tracks to enable the user to visualize them in Napari.

## 4.7 Stereocilia bundle height measurement

Hair cell stereocilia bundles are essential for hearing, as they convert acoustic vibrations into electrical signals that the brain detects as sound. Many deafness mutations cause bundle defects, including improper elongation. Accurately and consistently measuring stereocilia bundle heights in 3D images is therefore critical, but unfortunately laborious and costly, in particular for shorter bundles in either or both developing or mutant (e.g. *Eps8* KO mouse [37,54] cochlea tissues). Here we leverage our hair cell bundle segmentations to automate the measurement of bundle heights.
*Automated segmentation-derived method.* The steps for accurate and automated bundle height measurements involves calculating the distance from the tip to the base of the tallest row of stereocilia in hair cell bundles.

1. Iterate through each connected component in the labeled volume. Skip the background and any filtered components.
2. For each relevant component, identify all voxel coordinates that belong to the bundle.
3. Create a binary sub-volume for the current component where the component's voxels are marked as one, and all others are zeros.
4. Project the binary sub-volume along the z-axis to reduce it to a 2D projection by summing along the z-dimension and then applying a threshold to ensure the projection remains binary (values greater than 1 are set to 1).
5. Locate the highest (tip) and lowest (base) xy-coordinates in the 2D projection that have nonzero values:
   - Find the highest point by identifying the minimum xy-coordinate value that corresponds to 1 in the projection.
   - Find the bottom-most point by tracing downward from the centroid of the projection until reaching a xy-coordinate with a value of 0, then stepping back to the last non-zero coordinate.
6. Determine the z-coordinates of these points in the original 3D volume:

- For the highest point, find the z-coordinates where the voxel at the identified (x, y) location is 1.
- For the lowest point, set $z_{base}$ to the deepest slice index across the stack where the component is present.
7. Store the coordinates of the highest and lowest points. These are used for calculating the 3D Euclidean distance between the tip and the base of the bundle.
8. Calculate the distance using the Euclidean distance formula between the stored highest and lowest points.

$$\text{Distance} = \sqrt{(x_2 - x_1)^2 + (y_2 - y_1)^2 + (z_2 - z_1)^2}$$

where (x1, y1, z1) and (x2, y2, z2) are the coordinates of the first and second points in these dimensions. Note that a scaling factor should be used for each dimension to ensure the calculated distance accurately reflects the true spatial separation between the two points in real physical units.

Manual observer method (two observers): The height of manual stereocilia was measured in Fiji (ImageJ). For each bundle, an observer (i) identified the tallest stereocilium, (ii) placed a marker point at its base $(x, y, z)$, (iii) scrolled through the $z$-stack to the tip of the same stereocilium to place a second marker $(x, y, z)$, and (iv) calculated the 3D Euclidean distance between the base and tip using the known voxel dimensions (the same formula as above). Each bundle was independently measured by two observers.

## 4.8 Utilizing pre-trained ResNet50 for targeted classification of cochlear tonotopic regions

For training and validation, our dataset comprises 36 3D stacks from the BASE region containing 535 images, 35 3D stacks from the MIDDLE with 710 images, and 38 3D stacks from the APEX with 651 images. For testing the model, we used ten stacks from BASE, nine from MIDDLE, and ten from APEX that were withheld from the training data. The classification of each stack as BASE, MIDDLE, or APEX is determined through a majority voting mechanism applied across all frames of the associated stack, starting from the median and extending over a length of 13 frames.

In this study, we employed a modified ResNet50 model [51] using the PyTorch framework [55] to classify images of cochlear regions into three categories: BASE, MIDDLE, and APEX, which correspond to the high, middle, and low frequency response tonotopic positions of the cochlear spiral. The model, initialized with weights from pre-trained networks, was adapted to our specific task by altering the final fully connected layer to output three classes. When using a pre-trained ResNet50 model, the weights of the model have been adjusted based on its training on ImageNet [56], contains over 14 million images categorized into over 20,000 classes, where it has likely learned rich feature representations for a wide variety of images. This pre-training makes the model a strong starting point for most visual recognition tasks. To enhance model robustness and adaptability, training involved dynamic augmentation techniques including random resizing, cropping, flipping, color adjustments, and rotations, followed by normalization tailored to the ResNet50 architecture. This approach utilized both frozen and trainable layers, allowing for effective feature extraction adapted from pre-trained domain knowledge while refining the model to the specific needs of our dataset. Training was conducted over 100 epochs with real-time monitoring via TensorBoard, optimizing for accuracy through stochastic gradient descent with momentum. The best performing model was systematically saved to achieve marked improvements in classification accuracy.

## 4.9 IHC and OHC row classification

We implemented a deep learning strategy employing multi-class classification to precisely identify and categorize each hair cell bundle as either IHCs (first row from the buttom), OHC1 (second row from the buttom), OHC2 (third row from the buttom), and OHC3 (forth row from the buttom).

For the training and validation process, we utilized the same dataset used for the 3D segmentation. Each dataset was manually annotated into four rows: IHC, OHC1, OHC2, and OHC3. We employed a multi-class segmentation approach

using a U-Net architecture with a ResNet-50 backbone and a feature pyramid network. Transfer learning was utilized, leveraging pre-trained ImageNet weights to enhance model robustness. Given the existing 3D segmentation, our objective was simply to identify which cell corresponds to which row. To this end, we applied maximum projection to all frames in the stack to simplify and standardize the results.

Data augmentation has played a crucial role in enhancing the model's ability to generalize across different laboratory datasets, see Fig 21. Specifically, we have rotated the images by 10 and 20 degrees to both the right and left. Additionally, we have employed segmentation masks to mask out the raw images, effectively eliminating the background. This step creates additional training samples that focus on the foreground, helping the model to generalize better to other datasets where stains may not highlight the background, ensuring more accurate segmentation in diverse experimental conditions. Additionally, a CSV file that records distances is updated to include the new classifications, ensuring that each cell's identity (whether IHC, OHC1, OHC2, or OHC3) is documented.

## 4.10 Toward developing a foundational model for cochlea research: Integrating diverse data sources

For the final model available to the ear research community, we trained the architecture using a comprehensive dataset. This dataset included all the data from P5 (young mice), which consists of 45 stacks, as well as 10 stacks from P21 (adult mice), and 22 stacks provided by the Liberman lab. This amounted to a total of 901 2D images and 29,963 instances.

Given that this model is trained on young and adult mice of the same strain, as well as data from a different laboratory with varying settings, stains, and microscopy techniques, we consider this model to be the first trial in developing a foundational model. It is hoped that this model will work with data from other laboratories without requiring fine-tuning.

We are committed to maintaining our GitHub repository and actively encourage collaborators from other labs to share their data. By doing so, we aim to broaden the model's applicability and enhance its robustness, ultimately benefiting the entire ear research community.

## 4.11 Computational resources for all the experiments

All experiments were conducted on a local desktop computer equipped with a 13th Gen Intel(R) Core(TM) i9-13900K CPU, 128GB of RAM, and an NVIDIA GeForce RTX 4080 GPU with 16.0 GB of dedicated memory, running Microsoft Windows 11 Pro. We utilized the PyTorch framework [55] to implement all machine learning models to develop the Napari plugin. For training using the Facebook Research Detectron2 library [49], we utilized the Windows Subsystem for Linux (WSL), as the library is not supported natively on Windows. To integrate the algorithm into the Napari plugin [14], we built an executable that runs through one of the plugin's buttons via WSL. Consequently, users will need to set up WSL to utilize this plugin, details of which are thoroughly described in our GitHub documentation.

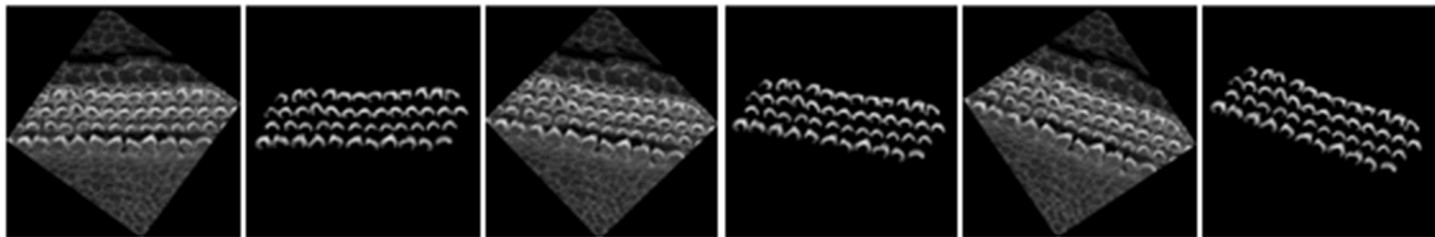

**Fig 21**. **Augmentation of maximum projection images from 3D confocal stacks for training a classification model that discriminates between IHC, OHC1, OHC2, and OHC3.**

## Supporting information

**S1 File. Describes the VASCilia workflow and features, user-guided adjustment of automated measurements, and the training procedures for stereocilia bundle segmentation.**
(PDF)

**S1 Fig. Representative P21 apex stereocilia segmentations (upright vs. flattened) shown across all $z$ planes with overlays and 3D/YZ views; masks follow stereocilia signal and exclude cuticular plate unless signal is present.**
(TIFF)

**S2 Fig. Full $z$-stacks (DS1–DS4) used to derive measurement crops; red boxes mark IHC/OHC positions.**
(TIFF)

**S3 Fig. Straight vs. rotated bundles (P5): raw stack, automatic segmentation, and 3D length readouts; 3D measurement handles both morphologies.**
(TIFF)

**S4 Fig. Angle computation on a PCP-deficit cochlear dataset; VASCilia closely matches manual Fiji measurements (paired $t$-test $p = 0.783$, Wilcoxon $p = 0.965$).**
(TIFF)

**S5 Fig. Representative images for $Cdh23^{+/-}$ and $Cdh23^{-/-}$ mice.**
(TIFF)

**S6 Fig. VASCilia screenshots with multiple samples (Lab A), set 1.**
(TIFF)

**S7 Fig. VASCilia screenshots with multiple samples (Lab A), set 2.**
(TIFF)

**S8 Fig. VASCilia screenshots with Lab B mouse cochlea data, set 1.**
(TIFF)

**S9 Fig. VASCilia screenshots with Lab B mouse cochlea data, set 2.**
(TIFF)

**S10 Fig. VASCilia screenshots with Lab B human cochlea data.**
(TIFF)

**S11 Fig. Base vs. apex in WT animals: base shows wider/stiffer bundles; apex finer/tighter bundles with reduced phalloidin intensity.**
(TIFF)

**S1 Table. Bundle height summary by genotype (WT/KO), tonotopic region (Base/Middle/Apex), and cell type (IHC/OHC); mean, SD, median, $N$.**
(PDF)

**S2 Table. Pairwise tonotopic contrasts in bundle height (Welch $t$-tests; Hedges' $g$; Holm-adjusted $p$); all significant except KO IHC Base–Middle.**
(PDF)

**S3 Table. Normalized fluorescence intensity by genotype/tonotopic region/cell type; mean, SD, median, *N*.**
(PDF)

**S4 Table. Pairwise tonotopic contrasts for normalized intensity with effect sizes and Holm-adjusted *p*-values.**
(PDF)

**S5 Table. Per-cell orientation angles (Manual vs. VASCilia), dataset 1; summary statistics.**
(PDF)

**S6 Table. Per-cell orientation angles (Manual vs. VASCilia), dataset 2; summary statistics.**
(PDF)

**S7 Table. Per-cell stereocilia lengths for *Eps8* KO (Manual vs. VASCilia; $n = 15$); no significant differences (paired *t* $p = 0.609$; Wilcoxon $p = 0.720$).**
(PDF)

**S8 Table. Per-cell stereocilia lengths for *Cdh23*$^{-/-}$ (Manual vs. VASCilia; $n = 15$); no significant differences (paired *t* $p = 0.851$; Wilcoxon $p = 0.639$).**
(PDF)

**S9 Table. Per-bundle orientation values for a PCP-deficit mouse dataset (Manual vs. VASCilia); tests indicate no significant difference.**
(PDF)

**S1 Data. Excel file containing all data related to Table 1, Table 2, Figs 5A–5D, Figs 7A–7B, Figs 10A–10B, Figs 11–15, and Figs 17A–17F. The name of each worksheet corresponds to the related figure or table.**
(XLSX)

## Acknowledgments

We would like to extend our sincere gratitude to the Liberman lab for generously sharing their data, to I-Hsun Wu (Research Development, Office of Research and Innovation, UC San Diego) for creating the illustrations in Fig 1 and Wendy Groves (Strategic Programs Team in the School of Biological Sciences at UC San Diego) for supervising I-Hsun Wu, and to Leo Andrade for providing the two SEM images of the stereocilia bundles of cochlear hair cells shown in Fig 1, to Cayla Miller for providing helpful feedback on the manuscript, and to Akanksha Biju (a student intern) for her assistance in annotating a subset of stereocilia bundles.

## Author contributions

**Conceptualization:** Yasmin M. Kassim, Uri Manor.

**Data curation:** Yasmin M. Kassim, David B. Rosenberg, Samprita Das, Xiaobo Wang, Zhuoling Huang, Samia Rahman, Ibraheem M. Al Shammaa, Samer Salim, Kevin Huang, Alma Renero, Yuzuru Ninoyu, Artur A. Indzhykulian.

**Formal analysis:** Yasmin M. Kassim.

**Funding acquisition:** Rick A. Friedman, Uri Manor.

**Investigation:** Yasmin M. Kassim, Uri Manor.

**Methodology:** Yasmin M. Kassim, David B. Rosenberg, Xiaobo Wang, Yuzuru Ninoyu.

**Project administration:** Uri Manor.

**Resources:** Xiaobo Wang, Rick A. Friedman, Artur A. Indzhykulian, Uri Manor.

**Software:** Yasmin M. Kassim.

**Supervision:** Yasmin M. Kassim, Rick A. Friedman, Uri Manor.

**Validation:** Yasmin M. Kassim, David B. Rosenberg, Samprita Das, Zhuoling Huang, Alma Renero, Artur A. Indzhykulian, Uri Manor.

**Visualization:** Yasmin M. Kassim, Samprita Das, Zhuoling Huang, Samia Rahman, Ibraheem M. Al Shammaa, Samer Salim, Kevin Huang, Alma Renero, Yuzuru Ninoyu.

**Writing – original draft:** Yasmin M. Kassim.

**Writing – review & editing:** Yasmin M. Kassim, Uri Manor.

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
