## [Editor Report · Decision Letter 0]

22 Apr 2025

Dear Dr Manor,

Thank you for submitting your manuscript entitled "VASCilia (Vision Analysis StereoCilia): A Napari Plugin for Deep Learning-Based 3D Analysis of Cochlear Hair Cell Stereocilia Bundles" for consideration as a Research Article by PLOS Biology. Please accept my sincere apologies for the delay in getting back to you last week as we consulted with an academic editor about your submission.

Your manuscript has now been evaluated by the PLOS Biology editorial staff, as well as by an academic editor with relevant expertise, and I am writing to let you know that we would like to send your submission out for external peer review.

Once your full submission is complete, your paper will undergo a series of checks in preparation for peer review. After your manuscript has passed the checks it will be sent out for review. To provide the metadata for your submission, please Login to Editorial Manager (https://www.editorialmanager.com/pbiology) within two working days, i.e. by Apr 24 2025 11:59PM.

Kind regards,

Richard

Richard Hodge, PhD

rhodge@plos.org

PLOS

---

## [Decision Letter · Decision Letter 1]

25 Jun 2025

Dear Dr Manor,

Thank you for your continued patience while your manuscript "VASCilia (Vision Analysis StereoCilia): A Napari Plugin for Deep Learning-Based 3D Analysis of Cochlear Hair Cell Stereocilia Bundles" was peer-reviewed at PLOS Biology. Please accept my sincere apologies for the delays that you have experienced during the peer review process. Your manuscript has now been evaluated by the PLOS Biology editors, an Academic Editor with relevant expertise, and by three independent reviewers.

In light of the reviews, which you will find at the end of this email, we would like to invite you to revise the work to thoroughly address the reviewers' reports.

As you can see, the reviewers think that the tool will be useful for the hair cell field, but they raise some important concerns that should be addressed in the revised version. Reviewer #1 notes that they have had problems installing the plugin and requests that more comprehensive installation instructions/tutorials are provided to ensure its user-friendliness and accessibility. The reviewer also raises concerns with the presentation of the paper and that it should be restructured to serve as a proof-of-principle for the different features. In addition, the reviewers several experimental revisions to fully demonstrate the utility and versatility of the approach. Reviewer #3 asks whether the tool can be adopted to study vestibular hair cells to broaden its impact and notes that additional validation data should be provided for the quantification of the stereocilia length. Finally, Reviewer #1 raises several technical concerns regarding the variability in the reported measurements.

Given the extent of revision needed, we cannot make a decision about publication until we have seen the revised manuscript and your response to the reviewers' comments. Your revised manuscript is likely to be sent for further evaluation by all or a subset of the reviewers.

**IMPORTANT - SUBMITTING YOUR REVISION**

*Re-submission Checklist*

*Published Peer Review*

*PLOS Data Policy*

*Blot and Gel Data Policy*

Best regards,

Richard

Richard Hodge, PhD

rhodge@plos.org

REVIEWS:

Reviewer #1: This would certainly be a useful tool for hair cell biologists and the paper gives a thorough account of its development and features. However, the paper is long and very dense and I think a lot of the text could be moved onto their website as files to read when installing or using the program. I think the paper should serve more as a proof of principle for the different features, however some of that kind of data is lacking, especially for the measurement features (Figs 8-13).

What I think could be improved is as follows:

1) I would have loved to test out this plugin but was not able to get it installed following instructions in the ReadMe document that they direct you to in the paper. I also could not access the other documents in the github site. In order for this to be more user-friendly there has to be better installation instructions provided in the paper including the system requirements needed on the computer. I know of one other person who eventually got it installed on their PC but was unable to run the example files they provide and the software was unable to open their confocal images without crashing. I think without clear user-friendly instructions for the installation this will not be very usable for most labs. Perhaps a detailed tutorial written for one of the demo files would be useful for guiding the use of the plug-in.

2) In Figure 4 (and the other figures), it is unclear to me how the bundle segmentations appear in 3D or how far they extend into the z plane. It seems that most of the images are fairly shallow stacks of flattened hair bundles and judging from Figure 4 (and from figure 28) the bundle segmentation is fairly shallow and uniform within the z-axis. What happens with images where the bundle is not flattened and varies in shape more throughout the z-axis? Also, does the segmentation extend into the cuticular plate at all or is there cell-to-cell variability in how much cuticular plate is included? The phalloidin intensities in the cuticular plate and the stereocilia can be quite similar at certain timepoints which may make segmentation of only the bundle more difficult. The segmentations in Figure 6 appear to be for more upright bundles. It would be good to show one of these close up at different rotations to see the segmentation at xz and yz planes and to keep the raw actin signal separate to see how the segmentation aligns with the bundle in all planes.

3) For the height measurements in Figure 8, how are the two observers performing the measurements? I'm assuming it's by marking the top and base of the tallest stereocilia within the z-stack? This approach to height measurement, especially given the varying orientation of these bundles, will likely add quite a lot of variability. Can you compare these measurements to another approach?

4) What are the four datasets in figure 8? Are they all from the same age and tonotopic region? The heights are averaged in Table 3 so I am assuming they are but the values for each dataset are quite different. If they are from the same area, then it appears the orientation of the stereocilia has a huge effect on the height measurement. This would make sense if the amount of stereocilia length in the z-axis rather than xy varies from cell to cell. The approach seems most viable only for the more flattened bundles.

5) The comments in the text about the time needed for height measurements in Fiji versus in the program does not seem accurate. It is likely that the user will have to go through and refine the base and top points that the program provides or at least would have to double check that they were accurate. This would significantly minimize the time difference.

6) While the measurements between the program and observers are not different (Table 3), they are all highly variable judging by the standard deviations in the table. This result likely derives from the variability in the bundle positions in the images and the way in which their calculation method intersects with this variability, which would apply to both program and observer calculations. The SD is about 25% of the mean length making finding significant lengths changes between conditions quite challenging. For the data in Figure 9a, one can see that the variability is quite large across all conditions and I'm not sure what level of significance could be measured. I would like to see a table of these values that include mean and standard deviation as well as any statistical tests.

7) The total intensity measurements in figures 9 and 10 are also incredibly variable. For Figure 9, I would also like to see a table of these values that include mean and standard deviation as well as any statistical tests. I don't think the authors can make any claims about the correlation between height and actin intensity without showing the statistics.

8) In figure 10, one can see some cell-to-cell variations in bundle segmentation that could contribute to intensity variation in the plot. Could you show the segmentation from another angle? It seems likely that there could be variation in the amount of cuticular plate actin that would be included in each segmentation which would affect these intensity measurements. How do these values compare to another approach for actin intensity measurement?

9) For figure 11, I am not convinced as to how useful this tool will be for other datasets from other ages. It seems like you would have to retrain it using your own images, but if you have already marked the tonotopic position of each image beforehand than what value does this feature provide?

10) For figure 12 and 13, could you compare your measurements to those collected using a different method or program?

Reviewer #2 (identities themselves as Bo Zhao): The study "VASCilia (Vision Analysis StereoCilia): A Napari Plugin for Deep Learning-Based 3D Analysis of Cochlear Hair Cell Stereocilia Bundles" by Kassim and colleagues developed a fantastic deep-learning based software capable of automatically and efficiently measuring many key cellular properties of hair cells including stereocilia length, actin content, and cell polarity. These features are critical for maintaining normal hearing function, and defects of them are known to contribute to hearing loss. Compared with manual measurements using tools such as Fiji, which are often time-consuming and labor-intensive, this automated approach offers a significant advancement. The reviewer greatly appreciates the valuable tools the team has brought to the field. To further strengthen this manuscript and potentially enhance the software's performance, I have the following suggestions.

1. The sentence in the abstract stating that "Using VASCilia, we found that the total actin content of stereocilia bundles (as measured by phalloidin staining) does not necessarily increase with bundle height, which is likely due to differences in stereocilia thickness and number" could be clarified further, as it currently lacks essential information such as the genotypes and developmental stages of the hair cells being referenced.

2. The fluorescence intensity measurements in Fig. 10 appear to show edge effects. Specifically, the intensity of OHCs in the upper left corner is much lower compared with those OHCs on the right side.

3. Would it be feasible to measure the height of the lower rows of stereocilia and quantify the height differences between different rows? Since different rows of stereocilia frequently overlap at the basal region, is there a way to identify the base of each stereocilium based on 3D images, such as through counterstaining with a stereocilia basal marker?

Several minors:

1. Please consider labeling all the three OHCs in Fig. 1 using the same color.

2. There is a typo in the last sentence of page 3.

3. The unit of length is missing in the Y-axis in Fig. 9a.

4. To improve clarity, please specify that it is the length of the tallest row of stereocilia in Fig. 9a.

Reviewer #3: This manuscript presented a very useful tool for studying the morphology of auditory hair cell stereocilia. I tried this plugin and the overall quality of it is solid, with tutorials and functions well documented. It has potential to be adopted by other labs to streamline hair bundle analysis. That said, there are still some questions about VASCillia that could be addressed for improving its clarity and utility. Please check the comments below:

Versatility of VASCillia: Considering that vestibular hair cells are widely used for studying PCP phenotypes, can VASCillia be easily adopted to study those hair cells with retraining the model?

3D contour of hair bundle: From the representative images provided by Figure 6, most of the reconstructed 3D contours seem to have a "plateau" at the top of the hair bundle, which may not accurately represent the true staircase-like structure of a hair bundle. I wonder if this could be improved by retraining the model. Or whether the tip regions are excluded by design that they are classified into pre-cellular zone.

Stereocilia length quantification: The automatic quantification of the stereocilia length is very powerful. However, it is important to note that VASCillia only quantifies the longest stereocilia within each bundle. Since VASCilia identifies the 3D contour of the hair bundle, I am curious if its function can be extended to infer the height of the second row of stereocilia by using the contour size of each z-section. This function could be very helpful to study the actin remodeling after noise damage or MET deficit. But I understand this could be very challenging to implement.

Validation of the stereocilia length quantification: It would be helpful to include human annotator ground truth of Esp8 KO mice in figure 9a. Also, I wonder if VASCilia can also provide accurate quantification for hair bundles with severe developmental deficit, such as MYO7A or CDH23 KO hair cells.

Fluorescence Intensity Measurements: Could the authors elaborate on the method used to normalize fluorescent signal intensity? For example, what are phalloidin intensities normalized to in figure 9b and 10? Are intensities calculated as total signal, or volume-normalized as signal/voxel? More detail on the normalization strategy would be helpful to minimize batch effects and replicate these analyses.

High-Throughput Computation of Bundle Orientation: It is a very useful function of quantifying the angle of stereocilia. I wonder if the author could provide some validation of cochlea with PCP deficit. Meanwhile, is it easy to quantify the V-shape angle of hair bundle?

Other comments: There are some missing definitions for the abbreviations, such as "FC" in Figure 2 convolutional neural network. Please include that in the manuscript for improving readability.

---

## [Decision Letter · Decision Letter 2]

21 Nov 2025

Dear Dr Manor,

Thank you for your patience while we considered your revised manuscript "VASCilia (Vision Analysis StereoCilia): A Napari Plugin for Deep Learning-Based 3D Analysis of Cochlear Hair Cell Stereocilia Bundles" for publication as a Methods and Resources Article at PLOS Biology. Please accept my sincere apologies for the delays that you have experienced during this round of the peer review process. This revised version of your manuscript has been evaluated by the PLOS Biology editors, the Academic Editor and the original reviewers.

Based on the reviews, we are likely to accept this manuscript for publication, provided you satisfactorily address the remaining points raised by the reviewers. After discussions with the academic editor, we will not make the additional validation of the intensity quantification methods essential for the revision, but we do ask that you make sure that the installation instructions are correct and that the program can run successfully, as this is essential to ensure that the tool will be accessible and useful for the field.

IMPORTANT

In addition, please make sure to address the following editorial and data-related requests that I have provided below (A-G):

(A) We routinely suggest changes to titles to ensure maximum accessibility for a broad, non-specialist readership. In this case, we would suggest a minor edit to the title, as follows. Please ensure you change both the manuscript file and the online submission system, as they need to match for final acceptance:

“VASCilia is an open-source, deep learning-based tool for 3D analysis of cochlear hair cell stereocilia Bundles"

(B) You may be aware of the PLOS Data Policy, which requires that all data be made available without restriction: http://journals.plos.org/plosbiology/s/data-availability. For more information, please also see this editorial: http://dx.doi.org/10.1371/journal.pbio.1001797

-Supplementary files (e.g., excel). Please ensure that all data files are uploaded as 'Supporting Information' and are invariably referred to (in the manuscript, figure legends, and the Description field when uploading your files) using the following format verbatim: S1 Data, S2 Data, etc. Multiple panels of a single or even several figures can be included as multiple sheets in one excel file that is saved using exactly the following convention: S1_Data.xlsx (using an underscore).

-Deposition in a publicly available repository. Please also provide the accession code or a reviewer link so that we may view your data before publication.

Figure 7A-B, 9A-B, 10, 15A-F

(C) Please also ensure that each of the relevant figure legends in your manuscript include information on *WHERE THE UNDERLYING DATA CAN BE FOUND*, and ensure your supplemental data file/s has a legend.

(D) Please note that we cannot accept sole deposition of code in GitHub, as this could be changed after publication. However, you can archive this version of your publicly available GitHub code to Zenodo. Once you do this, it will generate a DOI number, which you will need to provide in the Data Accessibility Statement (you are welcome to also provide the GitHub access information). See the process for doing this here: https://docs.github.com/en/repositories/archiving-a-github-repository/referencing-and-citing-content

(E) Please ensure that your Data Statement in the submission system accurately describes where your data can be found and is in final format, as it will be published as written there.

(F) Please ensure that you are using best practice for statistical reporting and data presentation. These are our guidelines https://journals.plos.org/plosbiology/s/best-practices-in-research-reporting#loc-statistical-reporting and a useful resource on data presentation https://journals.plos.org/plosbiology/article?id=10.1371/journal.pbio.1002128

- If you are reporting experiments where n ≤ 5, please plot each individual data point.

(G) Please note that per journal policy, the model system/species studied should be clearly stated in the abstract of your manuscript.

We expect to receive your revised manuscript within two weeks.

*Published Peer Review History*

*Press*

Best regards,

Richard

Richard Hodge, PhD

rhodge@plos.org

Reviewer remarks:

Reviewer #1: The authors have made some significant improvements to the manuscript in response to the reviewers' comments, although the length of the responses made the document difficult to read. We do appreciate that the authors provided a summary of the responses and then followed that with more details. Nevertheless, the manuscript is not an easy read.

Overall, we think the paper presents what could be a useful tool for some hair cell biologists and does a decent job of demonstrating its fidelity by comparing to other measurement techniques. Some of the specific use cases, like measuring hair bundle angles, should be very useful as the VASCilia approach allows analysis of many samples very quickly, allowing much later datasets that should have higher statistical power.

That said, this paper is in essence a methods paper and does not provide any new biological information, but rather uses case studies on some selected data sets to demonstrate each feature of the VASCilia software. As discussed below, the one case study where they claim a new biological result (reported in Fig. 9b) lacks validation and thus is difficult to accept. It is not also clear whether this tool will prove more useful than commercially available solutions (which admittedly will also require a lot of work to train and configure).

Thus this manuscript is an interesting presentation of a novel tool that may be useful to a subset of hair cell biologists. The tool is unlikely to be the only one used by those in the field, however, and there remain lingering questions about how easy it is to use for anyone trying to set it up in their laboratory.

Concerns and comments

1. A concern from the original review was over challenges in installing and using the program due to insufficient installation instructions and description of system requirements. The authors addressed these concerns and their new installation guide that includes system requirements seems much more thorough and easier to follow. Moreover, the text is certainly more streamlined now due to removing the user guides. Nevertheless, we were unable to get the program running ourselves because it apparently does not recognize the GPU we have in the computer. We recognize that in order to properly test the program we need to match the specifications laid out precisely, but it certainly limits the utility of the program if a new dedicated system must be purchased.

2. The authors are surely aware that there are commercial software programs (Arivis, Imaris, etc.) that, while not specifically designed for hair cells, already allow for 3D segmentation of hair bundles and include the ability to create deep-learning instance-based segmentation models (e.g., Arivis). However, these programs are quite expensive and having an open source program already trained for bundle segmentation will certainly be useful to hair cell biologists for certain analysis tasks. Nevertheless, we would appreciate some discussion of alternatives.

3. In the original review, we had concerns over the depth of segmentations in 3D, the possible inclusion of cuticular plate, and possible variance in its performance on flattened versus upright bundles. The authors addressed these concerns by providing a new supplementary figure that shows the segmentation with per slice overlays and orthogonal views. This figure is very helpful at providing a more thorough representation of what the segmentations look like in 3D relative to the actin signal as well as what they look like for a wider variety of bundles (in terms of both age and orientation (flat vs upright)). The changes to Fig. 10 as well as the addition of the Cdh23 -/- bundle case study were also helpful in showing the 3D segmentation performance more thoroughly for a wider variety of bundles.

4. For the height measurements, the authors have clarified our questions as to how the manual measurements were made, provided more details on the datasets used, and provided statistical comparisons for the results. I appreciate the new supplemental figures showing the z-stacks and the excel output for each cell as well as the new validation the KO models provided in tables S7 and S8. Overall, our concerns over the height measurements have been satisfied.

5. The use case for height measurement, bundle disorganization, and bundle orientation are the most successful because the authors include appropriate case studies using KO lines with known changes in the relevant measurement parameters and thoroughly compare their results to other measurement techniques (for height and orientation). However, the section on intensity measurements is the least successful in this regard. The authors should have used case study for intensity measurements that could be more easily validated and interpreted; without validation of their result (basal hair cells bundles more phalloidin intensity than apical hair bundles), it is impossible to know whether the results are correct. They certainly do not seem correct; we have examined thousands of cochlear hair cells labeled with phalloidin, and we are very surprised by the authors' claimed result. The only way that data in Fig. 9b should be included is if the result is corroborated by similar experiments with different markers (e.g., anti-actin antibody or antibody against a stereocilia cytoskeleton protein). As is, the data on total actin intensities in Fig. 9b are very hard to interpret and do not provide a good example about how these intensity measurements could be used more generally to give accurate and more high-throughput results than current measurement tools.

6. A different approach to use intensity measurements might be to use the bundle segmentations to measure the intensity of another protein expressed in hair bundles that they know would be missing or altered in expression in KO bundles (for instance, measure EPS8 expression in the WT and Eps8 KO bundles). The resulting data would be more interpretable and also easier to verify using standard measurement techniques.

Reviewer #2 (Bo Zhao, identifies himself): Thanks for developing this tool for the field.

Reviewer #3 (Sihan Li, identifies himself): I have one minor concerns about the new phalloidin quantification methods provided by the authors. 1. "summing restricted to the top (n) layers" may lead to a loss of information for hair bundles that has larger the Z-dimensions. 2. "padding shallower bundles by repeating the bundle's lowest value" might introduce false values that does not represent real phalloidin signal. However, it is nice to include these methods in VASCilia, which provide more options to meet different experimental needs.

Overall, the revised manuscript has addressed most of my previous questions. It also provides sufficient experimental validation and detailed descriptions of the analysis. I find the revisions satisfactory and would endorse the publication of this manuscript.

---

## [Editor Report · Decision Letter 3]

18 Dec 2025

Dear Uri,

On behalf of my colleagues and the Academic Editor, Alan Cheng, I am pleased to say that we can accept your manuscript for publication, provided you address any remaining formatting and reporting issues. These will be detailed in an email you should receive within 2-3 business days from our colleagues in the journal operations team; no action is required from you until then. Please note that we will not be able to formally accept your manuscript and schedule it for publication until you have completed any requested changes.

PRESS

Best wishes,;

Richard

Richard Hodge, PhD

rhodge@plos.org

PLOS
